# Size tuning of neural response variability in laminar circuits of macaque primary visual cortex

**Lauri Nurminen[†], Maryam Bijanzadeh[‡], Alessandra Angelucci***

Department of Ophthalmology and Visual Science, Moran Eye Institute, University of Utah, Salt Lake City, United States

**\*For correspondence:**
alessandra.angelucci@hsc.utah.edu

**Present address:** [†]College of Optometry, University of Houston, Houston, United States; [‡]Department of Neurology, UCSF Weill Institute for Neurosciences, University of California San Francisco, San Francisco, United States

**Competing interest:** The authors declare that no competing interests exist.

**Abstract** Surround suppression and neural response variability are widespread cortical phenomena thought to facilitate and impede, respectively, information processing and perception. Because manipulations that elicit neural response suppression often quench variability, it has been proposed that these two phenomena share a common origin. However, the relationship between surround suppression and variability has not been systematically examined. Surround suppression is mediated by multiple circuits and mechanisms that depend on the size of the sensory stimulus and cortical layer. Variability is also laminar dependent. To understand how surround suppression and variability interact and influence laminar processing, we used laminar electrophysiological recordings to examine how neural response variability and the shared variability among neurons are modulated by visual stimulus size across the layers of macaque primary visual cortex (V1). We find that surround suppression does not always quench variability. Instead, variability is tuned for stimulus size in a layer-dependent manner. In all layers, stimulation of the receptive field (RF) reduced both individual and shared variability relative to pre-stimulus baseline. Expanding the stimulus beyond the RF, into the near RF surround, further decreased variability in infragranular layers, but had little effect in granular and supragranular layers. In contrast, large stimuli extending into the far RF surround increased both individual and shared variability, relative to their value for a stimulus matched to the RF size, in supragranular layers, but decreased them or did not change them in granular and infragranular layers. Surprisingly, stimuli smaller than the RF could increase variability above baseline values, particularly in granular and infragranular layers. Our results indicate that surround suppression and variability are not governed by a single mechanism. Instead, multiple laminar-specific circuits and mechanisms shape variability, highlighting the need for revised models of neural response variability in cortical processing.

## Editor's evaluation

This valuable study examined relationships between stimulus size and response variability in the primary visual cortex of macaque monkeys. The authors present convincing evidence that, contrary to some previous reports, increases in stimulus size that induce surround suppression are not always accompanied by reductions in response variability across layers of visual cortex. These layer-specific patterns of response variability will help inform and constrain our understanding of functional circuitry in visual cortex.

## Introduction

Surround suppression and neural response variability are two widespread cortical phenomena thought to affect information processing and perception in seemingly opposite ways. Surround suppression

is a non-linear neural response transformation consisting of suppression of a neuron's response to a sensory stimulus presented inside its receptive field (RF) by a stimulus simultaneously presented in the RF surround (*Hubel and Wiesel, 1965*; *Blakemore and Tobin, 1972*; *Nelson and Frost, 1978*; *Sceniak et al., 2001*; *Cavanaugh et al., 2002*; *Levitt and Lund, 2002*; *Angelucci and Shushruth, 2013*). It is thought to facilitate information processing and the encoding of sensory stimuli, for example by facilitating visual perception (*Knierim and van Essen, 1992*; *Lamme, 1995*; *Nothdurft et al., 2000*), and reducing redundancies in sensory inputs (*Vinje and Gallant, 2000*; *Schwartz and Simoncelli, 2001*; *Haider et al., 2010*; *Pecka et al., 2014*). Neural response variability is the phenomenon in which repeated presentations of an identical visual stimulus produce neuronal spiking responses that differ from trial to trial (*Tomko and Crapper, 1974*; *Vogels et al., 1989*), and this trial-to-trial variability is correlated among neurons (*Cohen and Kohn, 2011*). Traditionally, neural response variability has been interpreted as 'noise' that impairs the fidelity of neural representations (*Shadlen and Newsome, 1998*; *Abbott and Dayan, 1999*; *Averbeck et al., 2006*; *Moreno-Bote et al., 2014*).

It has recently been hypothesized that response suppression (of which surround suppression represents a specific form) and neural variability are intimately connected and may share a common origin, as manipulations that elicit response suppression typically also quench variability (*Goris et al., 2024*). However, few studies have systematically examined the relationship between surround suppression and variability. *Festa et al., 2021* reported that on average across a population of V1 cells, stimuli approximately twice the diameter of the neurons' RF quench variability relative to variability measured for stimuli confined to the RF. However, in this study, individual V1 neurons showed a variety of effects of surround stimulation on variability, including increases, decreases, or no effect. A different study showed a decrease in correlated variability among V1 neurons following stimulation of the surround by large stimuli (12° diameter) (*Snyder et al., 2014*). However, experimental evidence indicates that surround suppression is mediated by multiple circuits and mechanisms, depending on stimulus size, spatio-temporal stimulus configuration, and cortical layer (*Webb et al., 2005*; *Angelucci et al., 2017*; *Bijanzadeh et al., 2018*; *Henry et al., 2020*). For example, small vs. large stimuli presented in the RF surround evoke distinct patterns of laminar activity (*Bijanzadeh et al., 2018*), consistent with the former engaging intrinsic horizontal connections, which extend into the 'near' RF surround of V1 neurons, and the latter corticocortical feedback connections, which are co-extensive with the 'far' RF surround of V1 neurons (*Angelucci et al., 2002*; *Angelucci et al., 2017*). As previous studies did not systematically address the relationship between stimulus size, single neuron variability, and correlated variability, it remains unclear whether near- and far-surround stimulation similarly affect variability, and whether their effects generalize to all cortical layers. Moreover, recent theoretical evidence has demonstrated that not all forms of variability are detrimental to encoding of sensory inputs and perception (*Averbeck et al., 2006*; *Ruff and Cohen, 2014*; *Ruff and Cohen, 2016b*; *Haefner et al., 2016*), and reports that variability is modulated by visual stimuli (*Churchland et al., 2010*; *Ruff and Cohen, 2016b*) have led some to suggest that variability may, instead, play a role in sensory information processing. For example, computational studies have assigned a role for variability in perceptual inference (*Fiser et al., 2010*; *Pouget et al., 2013*; *Orbán et al., 2016*; *Hénaff et al., 2020*), and suggested that cortical layers may play distinct roles in perceptual inference (*Bastos et al., 2012*). To obtain a comprehensive understanding of the relationship between surround suppression, variability, and laminar circuits, here we have used electrophysiological laminar recordings to examine how stimulus size affects changes in neural response variability and shared variability across the layers of macaque primary visual cortex (V1). Size tuning experiments allow us to isolate the driving feedforward thalamic inputs to the RF, from the modulatory inputs from the near and far RF surround, carried by intrinsic V1 and corticocortical feedback circuits, respectively (*Angelucci et al., 2002*; *Angelucci et al., 2017*; *Nurminen et al., 2018*). Understanding this relationship is important to constrain models of cortical circuits and the computations performed by networks of neurons.

We found that surround suppression does not always quench variability. Rather, both single neuron response variability and the shared variability among neurons are tuned for stimulus size, and this size tuning is layer dependent. In all layers, a stimulus matched to the RF size of the recorded neurons reduced both variability and shared variability compared to pre-stimulus baseline. However, modulation of variability by stimulation of the RF surround depended on both stimulus size and cortical layer. In infragranular (IG) layers, a stimulus involving the RF and the near surround decreased both response variability and shared variability, compared to their values for a stimulus matched to the RF

size. Instead, in supragranular (SG) and granular (G) layers, near-surround stimulation had small or no effects on variability. A larger stimulus extending farther into the far-surround increased variability, relative to its value for stimuli matched to the RF size, in the SG layers, but either did not affect or reduced variability in G and IG layers. Interestingly, we found that in a subset of neurons, particularly in G and IG layers, stimuli smaller than the RF could increase variability compared to pre-stimulus baseline. Our results point to multiple laminar-specific circuits and mechanisms as the source of variability, likely affecting cortical processing and perception in distinct ways, and calling for new models of neural response variability.

## Results

We recorded visually evoked local field potentials (LFP), single-unit (SU) and multi-unit (MU) spiking activity, using 24-channel linear electrode arrays (100 μm electrode spacing) inserted perpendicularly to the surface of area V1 in two sufentanil-anesthetized macaque monkeys (see Methods). For accurate assignment of recorded responses to cortical layers, verticality of the electrode array was verified by the spatial overlap and similarity of orientation preference of the MUs' minimum-response fields (mRFs) across the array, and confirmed by postmortem histology (as described in *Bijanzadeh et al., 2018*). Laminar boundaries were identified by current source density (CSD) analysis of LFP signals (*Mitzdorf, 1985*) averaged over all stimulus diameters used in this study. This allowed us to locate the G layer 4C as the site of the earliest current sink followed by a reversal to current source, the site of the reversal marking the bottom of the G layer; layers above and below G were defined as SG and IG, respectively. We used spiking activity in response to the same stimulus to identify the top and bottom of the cortex (*Bijanzadeh et al., 2018*).

To understand how stimulus size modulates cortical response variability across V1 layers, we measured Fano-factor and the shared variability among simultaneously recorded units as a function of grating diameter for 82 visually responsive MUs and 60 spike sorted, visually responsive SUs. At the beginning of each penetration, we mapped the mRFs of the recorded MUs (see *Bijanzadeh et al., 2018* for details). We next presented grating stimuli, centered on the aggregate mRF of the recorded MUs, to characterize the orientation, spatial frequency, and temporal frequency tuning of the recorded MU activity. We then selected the stimulus parameters that maximized the response of the majority of simultaneously recorded MUs across the array. Using these optimized parameters, we ran size tuning experiments in which the diameter of a drifting grating stimulus was varied from 0.1° (0.2° in one penetration) to 26°.

### Layer-dependent modulation of neural response variability by stimulus size

*Figure 1* shows size-tuning data for representative MUs recorded in SG, G, and IG layers. For all three example units, the Fano-factor decreased as the stimulus diameter was increased to fill the RF of the recorded neurons. However, for the SG layer MU (*Figure 1A*), increasing the stimulus diameter beyond the RF boundaries increased Fano-factor, relative to Fano-factor measured when the stimulus was matched to the size of the RF. In contrast, increasing the stimulus diameter beyond the RF boundaries did not affect Fano-factor for the G layer MU (*Figure 1B*). For the IG layer MU, increasing the stimulus diameter beyond the RF (here 0.5° diameter) led to a further decrease in Fano-factor when the stimulus was approximately twice the size of the RF (involving the 'near' RF surround), but for larger stimuli (involving the 'far' RF surround), Fano-factor increased back to approximately its value for a stimulus matched to the RF size (*Figure 1C*). Similar results were observed for the population of recorded MUs.

*Figure 2A* shows Fano-factor and mean firing rate, both normalized to their value evoked by a stimulus diameter equal to the RF diameter (see Methods for definition), averaged over the population of MUs in our sample, separately for the different layers. In SG (n = 31 MUs), G (n = 15 MUs), and IG (n = 36 MUs) layers, increasing the stimulus size up to the RF diameter progressively decreased Fano-factor and increased firing rate.

Large gratings extending into the RF surround of V1 cells are known to suppress the mean spiking response evoked by a stimulus confined to the cells' RF, a phenomenon known as surround suppression (*Sceniak et al., 2001*; *Angelucci et al., 2002*; *Cavanaugh et al., 2002*; *Levitt and Lund, 2002*;

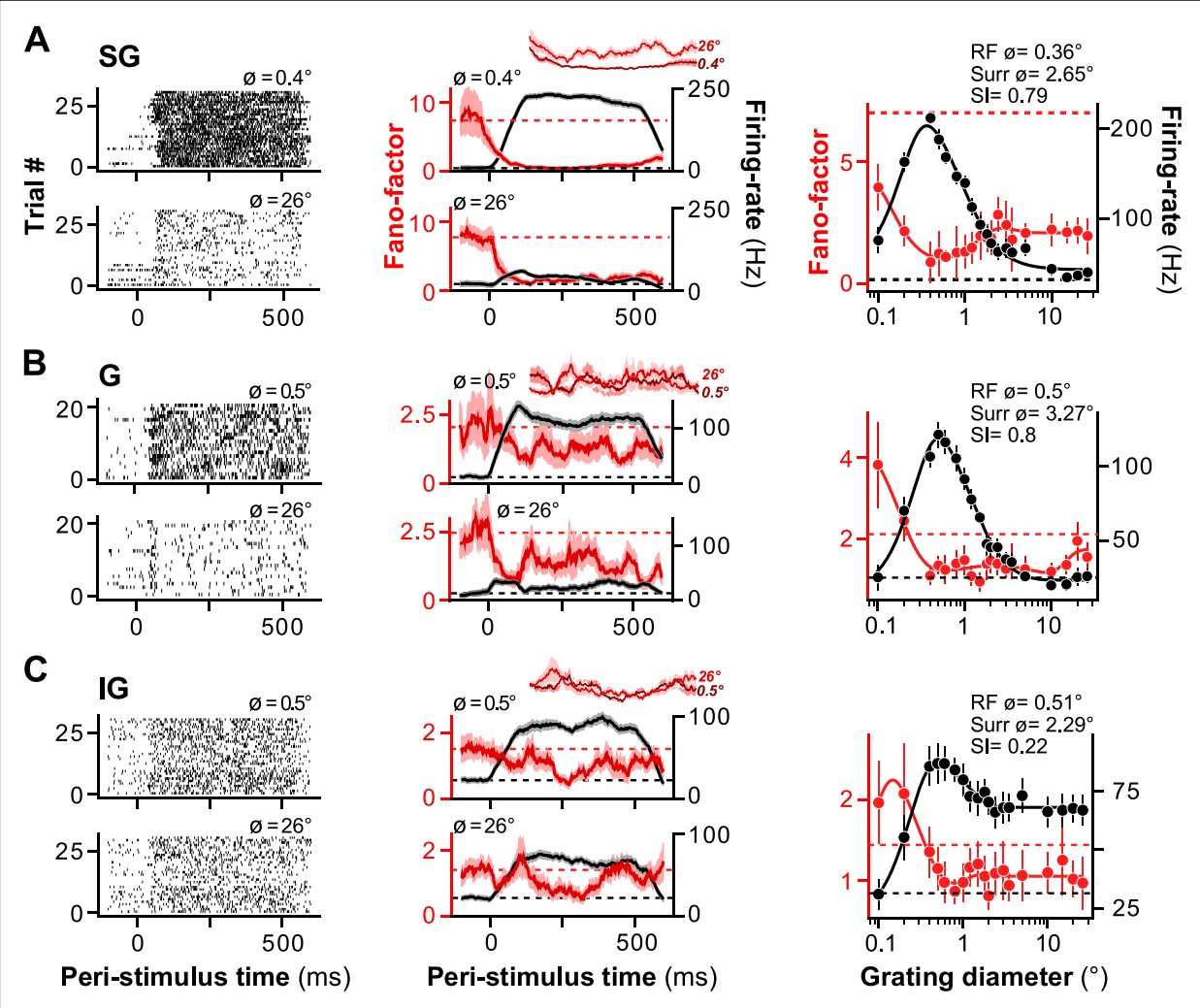

**Figure 1.** Size tuning of Fano-factor and mean firing rate in macaque V1: representative units. (**A**) Representative supragranular (SG) layer unit. **Left:** MU spike rasters measured at two stimulus diameters, either a diameter equal to the receptive field (RF) diameter of the recorded MU (top), or a diameter of 26° (bottom). **Middle:** Peri-stimulus time histograms (PSTHs) of Fano-factor (*red*) and mean firing rate (*black*) computed in a 100-ms rectangular sliding window for the same two stimulus diameters. The *shaded area* represents the standard deviation (s.d.) of the bootstrapped Fano-factor distribution (for the Fano-factor curve) or the standard-error-of-the-mean (s.e.m., for the firing-rate curve). *Inset*: Zoomed-in Fano-factor curves for the smaller (*darker red*) and larger (*lighter red*) stimulus diameters between 50 and 350 ms after stimulus onset. **Right:** Fano-factor (*red*) and firing rate (*black*) averaged over 50–350 ms after stimulus onset and plotted against the stimulus diameter. *Solid lines*: Fits to the data. *Dashed lines*: Baseline Fano-factor (*red*) and firing rate (*black*), measured prior to stimulus onset. Error bars are s.d. of the bootstrapped Fano-factor distribution (*red*) or s.e.m. (*black*). (**B**) Representative granular (G) layer unit. (**C**) Representative infragranular (IG) layer unit. Conventions in (**B, C**) are as in (**A**).

*Shushruth et al., 2009*; *Angelucci and Shushruth, 2013*). Consistent with these previous reports, as the stimulus size was increased beyond that of the RF diameter, firing rate decreased across all layers, and this suppression was strongest in the SG layers. However, increasing the stimulus diameter beyond the RF of the recorded MUs had different effects on Fano-factor in different layers. In SG and G layers, the minimum Fano-factor was reached for a stimulus the size (or near the size) of the RF diameter; as the stimulus diameter was increased beyond the RF diameter, Fano-factor significantly increased relative to its value for a stimulus matching the RF diameter in SG layers (*Figure 2A*, left), and while it appeared to increase also in the G layers, this increase was not statistically significant (*Figure 2A*, middle). In contrast, in IG layers, the minimum Fano-factor was reached at a stimulus diameter approximately twice the RF diameter (we term this the 'near surround'), and beyond this stimulus size, Fano-factor slowly increased back near its value for a stimulus matched to the RF

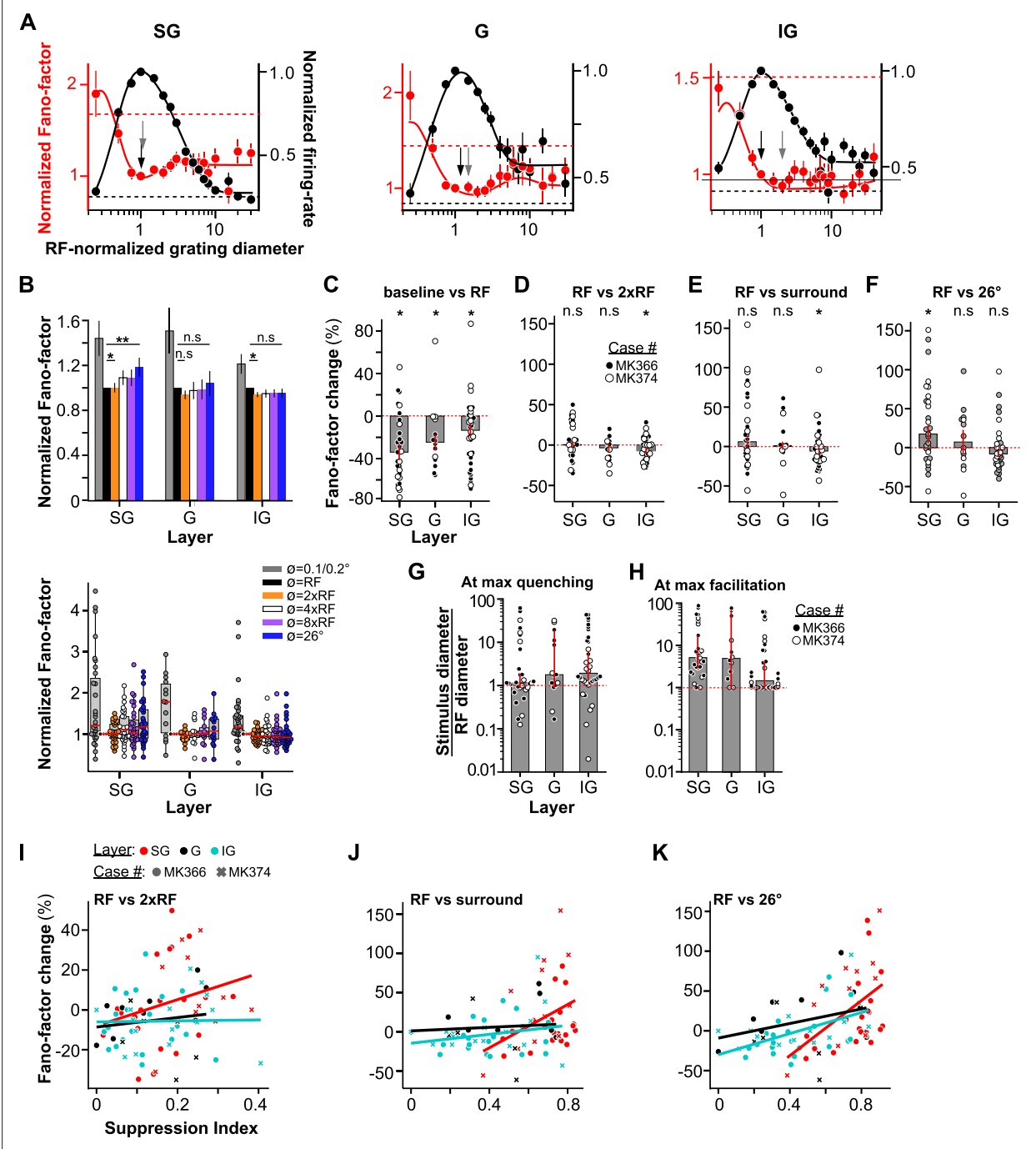

**Figure 2.** Size tuning of Fano-factor and mean firing rate: MU population data. (**A**) Average Fano-factor (*red*) and mean firing rate (*black*) as a function of stimulus diameter for the population of SG (**left**; *n* = 31), G (**middle**; *n* = 15), and IG (**right**; *n* = 36) layer MUs. *Dashed lines*: Average baseline Fano-factor (*red*) and firing rate (*black*); error bars: s.d. of the bootstrap distribution. Before averaging, stimulus diameter, Fano-factor, and firing rate of each multi-unit were normalized to the value at the receptive field (RF) diameter. In each panel, the *black arrow* points to the RF size, and the *gray arrow* to the stimulus size at minimum Fano-factor value. (**B**) **Top:** RF-normalized Fano-factor values averaged separately for six stimulus diameters: 0.1° or 0.2° (depending on which was the smallest diameter used for that penetration), a diameter equal to the RF diameter, a diameter twice the RF diameter (near surround), four times the RF diameter, eight times the RF diameter, and 26°. Error bars: s.d. of the bootstrap distribution. **Bottom:** Box plots of normalized Fano-factor for MUs in SG, G, and IG layers at the same six stimulus diameters as shown in the top panel. *Dots*: Fano-factor values for individual MUs; *red horizontal lines*: medians. The extent of the box marks the inter-quartile range of the data, and the whiskers mark the extent of the entire distribution, save for outliers that were outside 1.5 times the inter-quartile range. (**C**) Median percent change in Fano-factor relative to baseline induced by a stimulus matched in size to the RF diameter, for the different layers. *Dots*: Individual data points. Error bars (*red*): s.d. of the bootstrapped

*Figure 2 continued on next page*

*Figure 2 continued*

distribution. *Black and white dots* indicate data from different animals (case numbers indicated in panel D). (**D**) Median percent change in Fano-factor induced by a stimulus twice the RF diameter (near surround) relative to the Fano-factor value evoked by a stimulus matched to the RF diameter. Other conventions as in panel C. (**E**) Median percent change in Fano-factor induced by a stimulus with diameter equal to the surround diameter for individual cells (see Methods), relative to the Fano-factor value evoked by a stimulus matched to the RF diameter. Other conventions as in panel (**C**). (**F**) Median percent change in Fano-factor induced by a 26° diameter stimulus, relative to the Fano-factor value evoked by a stimulus matched to the RF diameter. Other conventions as in panel (**C**). (**G**) Median stimulus diameter at the smallest Fano-factor value (or max quenching), normalized to the RF diameter of each recorded MU, for different layers. Error bars: s.d. of the bootstrapped distributions. *Dots*: Individual MUs. (**H**) Median surround stimulus diameter at the highest Fano-factor value, normalized to the RF diameter. A value of 1 indicates that the highest Fano-factor value occurred at a stimulus size equal to the RF diameter. Error bars: s.d. of the bootstrapped distributions. *Dots*: Individual MUs. (**I**) Scatter plot of percent change in Fano-factor evoked by a stimulus with diameter twice the RF diameter relative to the Fano-factor evoked by a stimulus matched to the RF diameter vs. suppression index (see Methods). *Color dots* identify units in different layers, and the *different symbols* show data from two different animals, as indicated. *Lines* are regression lines fitted to the individual layer data. (**J**) Scatter plot of percent change in Fano-factor evoked by a stimulus with diameter equal to the *per-neuron* RF surround diameter relative to the Fano-factor evoked by a stimulus matched to the RF diameter vs. suppression index. Other conventions as in (**I**). (**K**) Percent change in Fano-factor evoked by a 26° diameter stimulus relative to the Fano-factor evoked by a stimulus matched to the RF diameter vs. suppression index. Other conventions as in (**I**).

The online version of this article includes the following figure supplement(s) for figure 2:

**Figure supplement 1.** Size tuning of Fano-factor across penetrations.

**Figure supplement 2.** Mean-matched Fano-factor analysis.

**Figure supplement 3.** Results for spike-sorted single units.

diameter (*Figure 2A*, right). The size tuning of the Fano-factor for individual penetrations is shown in *Figure 2—figure supplement 1A*.

To quantify these observations, *Figure 2B* shows Fano-factor estimated at six different stimulus diameters individually for each MU (values were extracted from functions fit to the data; see Methods), normalized to its value at a stimulus size matching the RF diameter, and then averaged (geometric mean) over all MUs. Consistent with the population size-tuning curves in *Figure 2A*, this analysis also showed a laminar dependence of the impact of surround stimulation on Fano-factor.

In SG layers, the normalized Fano-factor was lowest for a stimulus matching the RF diameter, and it began to increase for stimuli larger than about twice the RF diameter, reaching its maximum (relative to the value at the RF) at the largest stimulus size tested (26°); in SG layers, Fano-factor at 4xRF diameter, 8xRF diameter, and 26° diameter was significantly higher than Fano-factor at a stimulus size matching the RF diameter (one sample *t*-test, p < 0.05). Moreover, the Fano-factor for a stimulus twice the RF diameter was significantly higher than the value measured at the RF diameter (one sample *t*-test, p = 0.015).

In IG layers, the Fano-factor reached its minimum at a stimulus twice the RF diameter, that is extending into the near surround. Fano-factor at 2xRF diameter was significantly lower than Fano-factor for a stimulus matching the RF diameter (one sample *t*-test, p = 0.019) and Fano-factors at 4xRF diameter, 8xRF diameter, and 26° diameter were not significantly different from Fano-factor for a stimulus the size of the RF (one sample *t*-test, p > 0.45).

In G layers, the Fano-factor decreased as a small stimulus was increased to fill the RF size and did not change significantly for larger stimuli (one-sample *t*-test, p > 0.19). Fano-factor for a stimulus twice the RF size was also not significantly different from Fano-factor for a 26° diameter stimulus (p = 0.16, *t*-test).

*Figure 2C* plots for each layer the median percent change in Fano-factor, relative to baseline (pre-stimulus), evoked by a stimulus equal in size to the RF diameter of the recorded MUs. In all layers, presentation of visual stimuli in the RF significantly reduced Fano-factor relative to baseline (bootstrap test p < 0.03; median ± bootstrapped s.e. SG: –34.4 ± 7.55%, *n* = 31; G: –24.3 ± 7.84%, *n* = 14; IG: –13.7 ± 5.99%, *n* = 36). The median percent change in Fano-factor was significantly different in SG layers compared to IG layers (bootstrap test against the null hypothesis that the medians are equal: p = 0.024), but was not significantly different between the other layers (bootstrap test p > 0.25). The impact of near-surround (2xRF diameter) stimulation on Fano-factor was layer dependent. *Figure 2D* plots for each layer the median percent change in Fano-factor evoked by a stimulus matched to the near-surround of each MU, relative to a stimulus matched to the RF diameter. The dots in *Figure 2D* show this change for individual MUs. Whereas in SG and G layers, near-surround stimulation did not

significantly change Fano-factor, relative to its value for a stimulus matching the RF diameter, in IG layers, the percent change in Fano-factor caused by a stimulus in the near surround was significantly different from zero (bootstrap test p = 0.010). While near-surround stimulation in all layers could either increase or decrease Fano-factor in individual MUs, in IG layers, a greater fraction of the MUs showed percent changes in Fano-factor below zero. We also analyzed the effect on the Fano-factor of stimulating the per-neuron surround (*Figure 2E*, see Methods for definition). Only in the IG layers did the median Fano-factor change evoked by stimulating the per-neuron surround compared to the Fano-factor evoked by RF stimulation differ significantly from zero (bootstrap test, p = 0.01; SG and G layers: bootstrap test, p > 0.17). *Figure 2F* plots for each layer the percent change in Fano-factor evoked by a 26° stimulus (the largest we used) relative to a stimulus matched to the RF diameter. As for the previous analysis, the impact of far-surround stimulation on Fano-factor was layer dependent. In G and IG layers, there was no statistically significant change in Fano-factor (median ± bootstrapped s.e. G: 7.15 ± 15.6%, bootstrap test median >0: p = 0.44; IG: –7.99 ± 3.73%, bootstrap test median <0: p = 0.055). In contrast, in SG layers, there was a significant increase in Fano-factor as the stimulus involved the 'far surround' (SG: 17.4 ± 11.3%, bootstrap test median >0: p = 0.04). One MU with extreme change in Fano-factor (564%), caused by dividing by a number close to zero, was removed from the analysis. In summary, near-surround stimulation evoked a significant decrease in Fano-factor relative to its value for a stimulus matching the RF in IG layers, but no change in SG and G layers. Instead, far-surround stimulation evoked a significant increase in Fano-factor relative to its value for a stimulus matching the RF in SG layers, but no change in G and IG layers.

Importantly, in all layers, individual MUs showed a variety of changes, including increases, decreases, or no change in Fano-factor, regardless of whether the near surround, the per-neuron surround, or the entire far surround (26° diameter stimulus) was stimulated; thus, differences in median Fano-factor observed across layers reflected the relative proportion of one vs. another MU type. Thus, for example, the SG layers contained a larger proportion of MUs with far-surround-induced increases in Fano-factor than the other layers, but such units were present in all layers. This is demonstrated by the single MU plots shown in *Figure 2E, F*.

*Figure 2G* plots the median normalized stimulus diameter at the lowest Fano-factor value (i.e. at max Fano-factor quenching) in different layers, and the dots indicate this value for each individual MU. Median stimulus diameter at max Fano-factor quenching normalized to the RF diameter was 1.14 (lower and upper bound of the 99% bootstrapped confidence interval of the median: 0.90, 2.42) in SG layers, 1.80 (0.91 19.41) in G layers, and 1.90 (1.32 6.21) in IG layers. The same data, color coded, for individual penetrations are shown in *Figure 2—figure supplement 1B*.

*Figure 2H* plots the smallest surround stimulus diameter at max Fano-factor value. For example, for a unit for which surround stimuli of any size reduced Fano-factor or did not change it relative to its value for a stimulus matched to the RF, the stimulus size at max Fano-factor value would correspond to the RF diameter. For SG and G layers, max Fano-factor value occurred when the stimulus diameter was approximately five times the RF diameter [median stimulus diameter normalized to the RF diameter and lower and upper bounds of bootstrapped 99% confidence interval, SG: 5.02 (2.93, 32.1); G: 4.84 (1.0, 67.2)]. In contrast, in IG layers, max Fano-factor value occurred at a stimulus diameter similar to the RF diameter [1.46 (1.0, 4.13)]. The same data, color coded, for individual penetrations are shown in *Figure 2—figure supplement 1C*. These data demonstrate that an increase in Fano-factor for stimuli larger than the RF, particularly prominent in the SG layers, is seen in cells across all penetrations.

To better understand the mechanisms that underlie the effects of surround stimulation on Fano-factor, we investigated whether the surround-induced percent change in Fano-factor linearly depends on the strength of surround suppression. We plotted percent Fano-factor change against suppression index (see Methods) and computed the Pearson's correlation between the two (*Figure 2I–K*). For near-surround stimulation, the correlation between percent Fano-factor change and suppression index was not significantly different from zero in any of the layers (SG: $r = 0.25$; G: $r = 0.15$; IG: $r = 0.02$; p > 0.16; *Figure 2I*). When the per-neuron surround was stimulated, the correlation was significant in SG layers ($r = 0.37$, p = 0.04), but not in G or IG layers ($r = 0.08$ -G-, $r = 0.22$ -IG-, p > 0.2; *Figure 2J*). For far-surround stimulation with a 26° diameter stimulus, the correlation was significant in SG and IG ($r = 0.48$ -SG-, $r = 0.50$ -IG-, p < 0.006), but not G ($r = 0.30$, p = 0.30), layers (*Figure 2K*). In summary, strength of surround suppression was positively correlated with percent change in Fano-factor only for far-surround stimulation in SG and IG layers. We also found that surround suppression was stronger

in the MUs in which Fano-factor was increased by a 26° diameter grating relative to Fano-factor measured in the RF (one-way ANOVA, strength of surround suppression conditioned on whether RF surround increased, decreased, or had no effect on Fano-factor relative to RF stimulation, p = 0.001, n = 82). Specifically, for the MUs in which surround stimulation increased Fano-factor relative to RF stimulation (n = 21), the strength of surround suppression was 74.1 ± 2.91%, while it averaged 54.7 ± 3.47% for the MUs in which surround stimulation did not affect variability (n = 49), and 51.8 ± 5.63% for the MUs in which surround stimulation reduced variability (n = 12).

In all layers, statistically significant increases in variability (as determined by bootstrapping; see Methods) induced by surround stimulation had larger magnitude than decreases in variability (increase vs. decrease SG: 103 ± 18.9% vs. −36.5 ± 6.10%; G: 68.1 ± 14.4% vs. −40.1 ± 20%; IG: 91.6 ± 31.0% vs. −36.5 ± 6.09% independent samples *t*-test pooled over layers and computed over the absolute value of the Fano-factor change induced by RF surround, p = 0.002).

To rule out that changes in Fano-factor with stimulus size are trivially related to changes in firing rate, we performed a 'mean-matched' analysis (*Mitchell et al., 2009*; *Churchland et al., 2010*) (see Supplementary Methods, Supplementary Results, and *Figure 2—figure supplement 2*). This analysis showed similar results as described above for the non-mean-matched analysis. One difference between the two analyses was that the mean-matched analysis demonstrated a small but significant decrease in Fano-factor for near-surround stimulation, relative to its value for RF stimulation, also in the G layers (vs. no change in Fano-factor for the non-mean-matched analysis; *Figure 2B, D*). Moreover, the mean-matched analysis demonstrated a significant decrease in Fano-factor for far-surround stimuli relative to Fano-factor for a stimulus matched to the RF in both the G and IG layers (see Supplementary Results), while there was no difference in Fano-factor between these two stimulus sizes in these layers for the non-mean-matched analysis (*Figure 2B, F*).

We also replicated this analysis on a smaller population of spike-sorted SUs from SG and IG layers (see Supplementary Methods, Supplementary Results, and *Figure 2—figure supplement 3*). Similar to the MU analysis, the SU analysis demonstrated a diversity of responses in all layers (*Figure 2—figure supplement 3A*). As in MUs, a larger fraction of SUs in the SG layers than in the IG layers showed max Fano-factor facilitation for stimuli larger than the RF size (*Figure 2—figure supplement 3D*). In *Figure 2—figure supplement 3B*, however, this increase in Fano-factor for large stimuli in the SG layers did not reach statistical significance, due to the small SU sample size. The analysis in panel D, however, is more sensitive, because it does not perform measurements at predefined stimulus sizes (as in panel B), but searches for Fano-factor increases at any stimulus size for each individual cell.

A larger fraction of SUs in both SG and IG layers showed a decrease in Fano-factor for near-surround stimuli, relative to Fano-factor for a stimulus matched to the RF diameter (*Figure 2—figure supplement 3B, C*); this was in contrast to the MU data in which this decrease in Fano-factor for near-surround stimulation was only observed in the G and IG layers (*Figure 2B, D, G*). This was because our larger population of MUs in SG layers contained a larger fraction of individual MUs with lowest Fano-factor value for stimuli matched to the RF size, compared to our smaller population of SUs.

## Amplification of cortical response variability by small stimuli

It has been previously reported that the onset of a visual stimulus reduces cortical response variability relative to pre-stimulus baseline (*Churchland et al., 2010*). However, previous studies used relatively large stimuli, and the impact of stimulus size on response variability has not been explored. Previous experimental studies (*Ichida et al., 2007*) have shown, and several models of cortical dynamics predicted, that when the cortex is weakly driven (e.g. by a small stimulus), the cortical state is dominated by excitation, whereas it is dominated by inhibition when the cortex is strongly driven, for example, by a large stimulus (*Schwabe et al., 2006*; *Schwabe et al., 2010*; *Rubin et al., 2015*; *Hennequin et al., 2018*). In an excitation-dominated cortical state, stochastic supralinear stabilized networks predict amplification of response variability relative to pre-stimulus baseline (*Hennequin et al., 2018*). To test this model's prediction, we examined the impact of small stimuli on response variability.

*Figure 3A* shows the response of three example MUs to small gratings (0.1–0.6° diameter) centered on the RF. In all three MUs, this small stimulus-evoked firing rates and Fano-factors values higher than those measured during the pre-stimulus baseline.

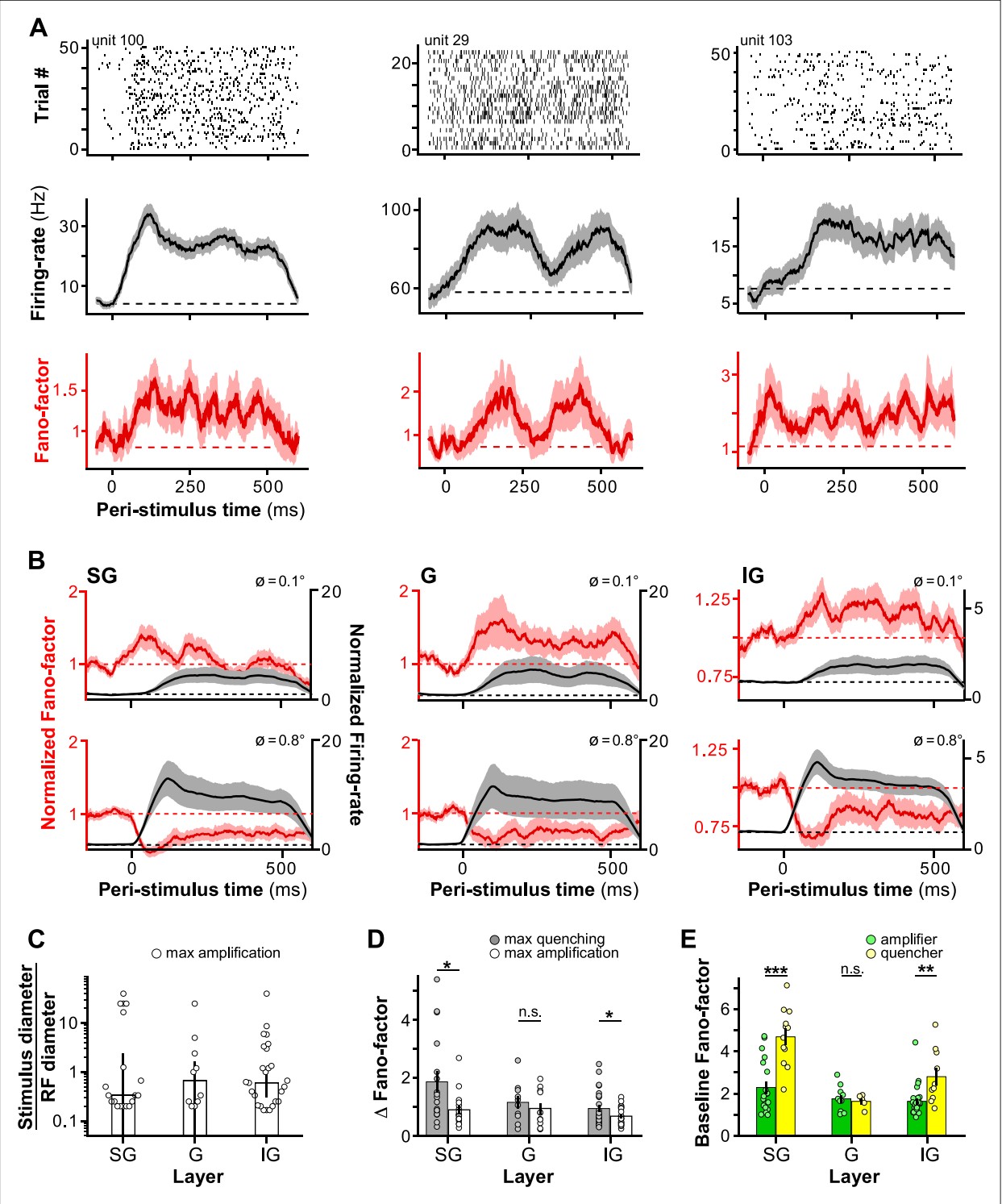

**Figure 3.** Amplification of cortical response variability by small visual stimuli. (**A**) Three example MUs showing stimulus-evoked increases in firing rate and Fano-factor relative to baseline for small stimuli. **Top:** Spike rasters. **Middle:** PSTH of firing rate. *Dashed line*: Baseline firing rate. **Bottom:** PSTH of Fano-factor. *Dashed line*: Baseline Fano-factor. (**B**) Population-averaged time course of Fano-factor (*red*) and firing rate (*black*) in SG (**left**), G (**middle**), and IG (**right**) layers computed at two stimulus diameters (**top:** 0.1°, **bottom:** 0.8°). Both the Fano-factor and firing rate were normalized to the pre-stimulus baseline of each unit before averaging. (**C**) Median stimulus diameter evoking the largest magnitude increase in Fano-factor, normalized to the receptive field (RF) diameter of the recorded MUs, for different layers. Error bars: s.d. of the bootstrapped distributions. *Dots here and in* (**D, E**) individual MU data. (**D**) Median difference in Fano-factor (Fano-factor at the stimulus diameter causing the largest change in Fano-factor minus the

*Figure 3 continued on next page*

*Figure 3 continued*

baseline Fano-factor) ± s.d. of the bootstrapped distributions at max quenching (*gray*) and max amplification (*white*) for different layers. (**E**) Mean baseline Fano-factor for amplifier (*green*) and quencher (*yellow*) MUs. Error bars: s.e.m.

The online version of this article includes the following figure supplement(s) for figure 3:

**Figure supplement 1.** Variability amplification is not due to eye movements.

Amplification of variability for small stimuli was seen also at the population level. *Figure 3B* compares Fano-factor evoked by a 0.1° diameter grating with that evoked by a grating of diameter equal to the RF diameter of the recorded MUs, normalized to the pre-stimulus baseline, and averaged over the population of SG ($n$ = 31, left), G ($n$ = 15, middle), and IG ($n$ = 36, right) MUs. Presentation of the small stimulus significantly increased Fano-factor relative to pre-stimulus baseline in G and IG ($p <$ 0.05, one-sample $t$-test, $n$ = 15 and 36, respectively), but not SG ($p$ = 0.14, $n$ = 31), layers. Consistent with previous studies (*Churchland et al., 2010*), in all layers, the larger stimulus instead decreased Fano-factor relative to baseline. There was no obvious difference in firing rates across the layers that could have explained the increase in Fano-factor in G and IG layers for smaller stimuli, but not in SG layers.

To provide a better understanding of variability amplification across layers, we performed an MU-by-MU analysis. This revealed significant variability amplification for small stimuli in all layers. We included in this analysis only MUs showing statistically significant increase and decrease (see Methods) in Fano-factor relative to baseline for at least one data point. While all MUs in our sample showed statistically significant stimulus-evoked decreases in Fano-factor, 67% of these units (55/82) also showed statistically significant increases in Fano-factor for presentation of small stimuli. The proportion of MUs showing both increases and decreases in stimulus-evoked Fano-factors was fairly constant across layers (**SG**, 61%; **G**, 67%; **IG**, 72%).

On average, the largest stimulus-evoked increase in Fano-factor for the cells that showed variability amplification relative to baseline was observed for stimulus diameters smaller than the RF of the recorded MU (*Figure 3C*; median stimulus diameter normalized to the RF diameter at the largest increase in Fano-factor relative to baseline ± s.d. of the bootstrapped distribution: **SG**, 0.33 ± 1.20; **G**, 0.67 ± 0.52; **IG**, 0.63 ± 0.26). We found that stimulus-evoked increases in Fano-factor were smaller in magnitude than stimulus-evoked decreases in Fano-factor. The difference between the magnitude of maximum variability quenching and magnitude of maximum variability amplification was statistically significant in SG and IG layers (*Figure 3D*; mean ± s.e.m. quenching vs. amplification: **SG**, 1.85 ± 0.37 vs. 0.89 ± 0.14, $t$-test, $p$ = 0.01; **IG**, 0.93 ± 0.12 vs. 0.67 ± 0.06, $p$ = 0.03), but not in G layers (**G**, 1.14 ± 0.21 vs. 0.93 ± 0.19, $p$ = 0.23). For this analysis, we removed outlier data points that were at least 2.5 absolute median deviations above or below the median.

Across the entire population, the MUs showing variability amplification for small stimuli (here termed 'amplifiers') had a significantly lower baseline Fano-factor than the MUs in which a stimulus always reduced variability (termed 'quenchers') (mean baseline Fano-factor ± s.e.m.: 3.44 ± 0.33 for quenchers vs. 1.90 ± 0.13 for amplifiers, $t$-test, $p$ = 0.000001). However, this varied by layer (*Figure 3E*); in SG and IG layers, baseline Fano-factor was significantly higher for the quenchers than for the amplifiers (mean baseline Fano-factor ± s.e.m.: **SG**, quenchers 4.70 ± 0.39, $n$ = 12, vs. amplifiers 2.30 ± 0.29, $n$ = 19, $t$-test, $p$ = 0.00023; **IG**, quenchers 2.82 ± 0.40, $n$ = 10, vs. amplifiers 1.65 ± 0.14, $n$ = 26, $t$-test, $p$ = 0.0013). Instead, in the **G** layer, baseline Fano-factor did not differ significantly between these two groups (quenchers 1.65 ± 0.14, $n$ = 5, vs. amplifiers 1.77 ± 0.19, $n$ = 10, $t$-test, $p$ = 0.68). Moreover, baseline firing rate was significantly lower in the amplifiers compared to the quenchers (mean ± s.e.m. baseline firing rate: 4.1 ± 0.4 vs. 6.0 ± 0.7 Hz, $t$-test, $p$ = $2^{-12}$). Importantly, however, all amplifiers also showed variability quenching for larger stimuli. This suggests that a floor effect, for example, due to low baseline firing rates, cannot explain the variability amplification in our data (see Discussion). Variability amplification for small stimuli could potentially arise as a result of slow drifting eye movements which can occur under anesthesia. However, we controlled for eye movements by frequent remapping of the mRF location (see Methods). *Figure 3—figure supplement 1* rules out eye movements as a potential cause of increased variability for stimuli smaller than the RF size. This figure shows an example penetration in which amplification of variability for small stimuli was seen in 12 MUs, and this amplification could not be explained by eye movements. Because our penetrations were nearly perfectly vertical to the cortical surface (as assessed by alignment of RF location and

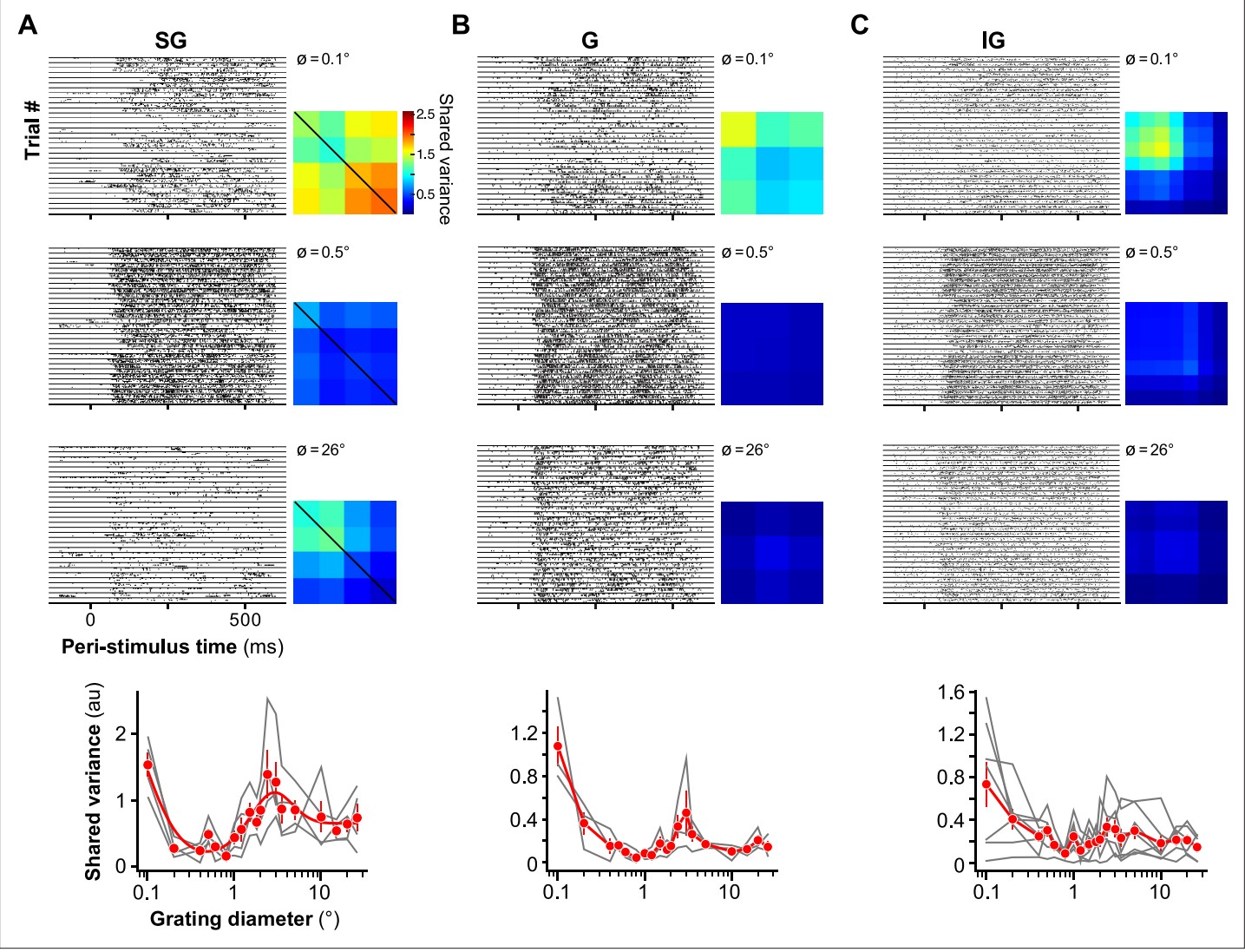

**Figure 4.** Size tuning of shared variance across V1 layers: example penetration. (**A**) Left: Raster plots showing the spike times of four simultaneously recorded SG MUs, across several trials, in response to 0.1° (top), 0.5° (middle), and 26° (bottom) diameter gratings. The responses of all four neurons in a single trial are shown between two consecutive horizontal lines. Horizontal lines separate different trials. **Right:** Network covariance matrices estimated with a single-factor Factor Analysis for each of the same three different stimulus diameters. The diagonal of the network covariance matrix holds the shared variance for each recorded MU. **Bottom:** Shared variance as a function of stimulus diameter. The *red markers* show mean ± s.e.m. of the shared variance computed for the SG MU population recorded in this example penetration (*n* = 4). The gray curves show the data for the four individual MUs. (**B, C**) same as in (**A**), but for G (*n* = 3) and IG (*n* = 7) layer MUs, respectively.

similarity of tuning functions; see *Bijanzadeh et al., 2018* for quantification), eye movements would be expected to cause a decrease in firing rate in some trials across the entire cortical depth, that is across all contacts in a penetration; this was not observed in the penetration shown in *Figure 3— figure supplement 1*.

## Layer-dependent size tuning of network variability

The results presented above indicate that the response variability of visual cortical neurons is modulated by stimulus size. However, the impact of neural response variability on visual processing also depends on how strongly variability is shared across neurons (*Shadlen and Newsome, 1998*; *Bair et al., 2001*). To determine the impact of stimulus size on shared variability, we exploited the covariance of simultaneous recordings obtained with electrode arrays.

The left panels in *Figure 4A–C* show raster plots of spiking activity of simultaneously recorded MUs in a single example penetration spanning all layers (SG, *n* = 4 units, *Figure 4A*; G, *n* = 3, *Figure 4B*; IG, *n* = 7, *Figure 4C*), in response to presentation of gratings of three different diameters (0.1° top, 0.5° middle, 26° bottom). Qualitative inspection of these raster plots reveals that, following presentation of the 0.1° stimulus, the responses of the entire neuronal population waxed and waned in unison. In some trials, all MUs spiked vigorously, whereas in other trials, the entire population was silent. However, the population responses appeared less coordinated following presentation of the 0.5° diameter grating. The covariation in population responses to presentation of the 26° diameter grating, instead, appeared to be layer dependent. In all layers, responses to a 26° stimulus were reduced compared to those to a 0.5° diameter grating. However, compared to the responses to the 0.5° stimulus, neural activity in response to the 26° stimulus appeared more strongly coordinated in SG layers but remained relatively uncoordinated in G and IG layers.

To quantify these observations, we used Factor Analysis (see Methods). This allowed us to isolate the network variability (i.e. the variability that is shared across MUs) from the private spiking variability (i.e. the variability that is private to each recorded MU) (*Churchland et al., 2010*). Factor analysis decomposes the measured covariance matrix into a low-rank network covariance matrix and a diagonal matrix that holds the private variances for each MU. Using cross-validated leave-unit-out predictions, we determined that a single-latent dimension was sufficient to model the current data; therefore, we modeled the network covariance matrix with a single factor and took the diagonal of the network covariance matrix as the shared variance (*Churchland et al., 2010*). A separate single-factor model was learned for each layer, stimulus condition, and penetration. Importantly, the results of this analysis do not imply that V1 activity is one-dimensional, but rather reflect the sampled V1 activity. Because we used linear arrays with 100 µm contact spacing, and we factored our analyses based on layer, our sample consists of nearby neurons, at most 700 µm apart. Moreover, to maximize the similarity of the RFs of the recorded neurons, we ensured that the recording probe was inserted vertically into the cortex; this was determined based on the overlap in RF locations and similarity of the orientation preferences of neurons recorded in a single penetration (for an example penetration, see Figure 1 in *Bijanzadeh et al., 2018*).

The right panels in *Figure 4A–C* show the network covariance matrices for the same MUs and three stimulus diameters used for the raster plots. In all layers, shared variance was highest when the smallest of the three stimuli was presented and dropped to near zero in response to the 0.5° stimulus (0.1° vs. 0.5° mean ± s.e.m.: SG, 1.53 ± 0.18 vs. 0.48 ± 0.08, *n* = 4; G, 1.07 ± 0.18 vs. 0.16 ± 0.02, *n* = 3; IG, 0.74 ± 0.21 vs. 0.30 ± 0.06, *n* = 7). The impact of larger stimuli on shared variance, instead, depended on layer. Compared to shared variance in response to the 0.5° stimulus, shared variance in response to the 26° stimulus increased in SG layers (0.5° vs. 26° mean ± s.e.m.: 0.48 ± 0.08 vs. 0.74 ± 0.20, *n*=4), did not change in G layers (0.16 ± 0.02 vs. 0.14 ± 0.05, *n* = 3), and decreased in IG layers (0.30 ± 0.06 vs. 0.14 ± 0.04, *n* = 7). The bottom panel of *Figure 4A–C* shows, for the same example MUs, shared variance as a function of stimulus diameter for all diameters used in our study (0.1–26°).

*Figure 5A* shows size tuning of firing rate and shared variance for the population of MUs recorded in SG, G, and IG layers. Shared variance was tuned for stimulus size in a manner that resembled the size tuning of Fano-factor (compare with *Figure 2A*). In all layers, increasing the stimulus diameter from 0.1° to a size equal to the aggregate RF diameter of the recorded MUs progressively increased firing rate but decreased shared variance (*Figure 5A, B*). Shared variance also decreased relative to baseline for a stimulus matched to the RF diameter; the percent change in shared variance was similar across layers and there were no obvious differences across the two animals used in this study (*Figure 5C*; median percent change in shared variance ± s.d. of the bootstrap distribution, SG: –47.5 ± 11.2%, G: –29.6 ± 18.9%, IG –31.9 ± 17.7). The median percent change in shared variance was significantly below zero in all layers (one-tailed bootstrap test, p < 0.02). In contrast, the effect on shared variance of stimulating the RF with larger stimuli extending beyond the RF, into the RF surround, differed across layers. Near-surround stimulation (2xRF diameter) significantly reduced shared variance compared to its value for a stimulus matched to the RF, only in IG layers, but not in G and SG layers (*Figure 5D*; median percent change in shared variance ± s.d. of the bootstrapped distribution, SG: –4.60 ± 9.58%, G: –8.31 ± 11.11%, p ≥ 0.21, IG: –28.8 ± 7.01%, p < 0.015). Stimulating the per-neuron surround did not affect the shared variance in any layer (*Figure 5E*; median percent change in shared variance ± s.d. of the bootstrap distribution, SG: 4.55 ± 20.6%, G: 10.1 ± 20.2%, –37.4 ±

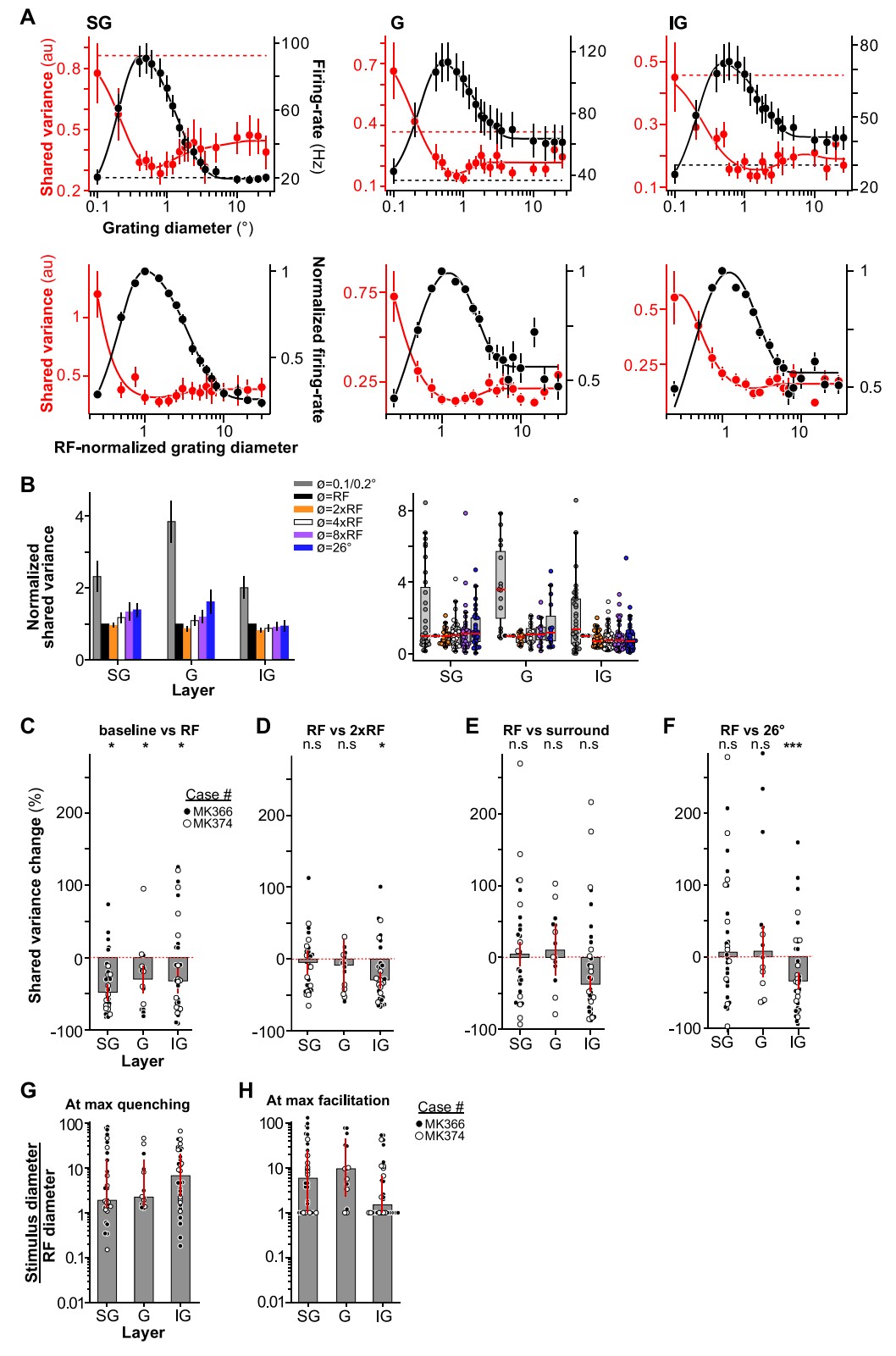

**Figure 5.** Size tuning of shared variance across V1 layers. (**A**) **Top:** Size tuning of shared variance (*red*) and firing rate (*black*) averaged over the population of SG (left), G (middle), and IG (right) MUs. *Solid lines*: Fits to the data. *Dashed lines*: Baseline shared variance (*red*) and firing rate (*black*), measured prior to stimulus onset. Error bars are s.d. of the bootstrapped shared variance distribution (*red*) or s.e.m. (*black*). **Bottom:** Mean firing rate (*black*)

*Figure 5 continued*

and mean shared variance (*red*) as a function of normalized stimulus diameter, averaged over the population of recorded MUs separately for the different layers (SG, *n* = 31 units; G, *n* = 15; IG, *n* = 36). The stimulus diameter was normalized to the receptive field (RF) diameter. (**B**)  **Left:** RF-normalized shared variance averaged over the population at six different stimulus diameters (as indicated); shared variance values at specific stimulus sizes were extracted from functions fitted to the size-tuning data (see Methods). Error bars: s.d. of the bootstrapped distribution. **Right:** Box plots of normalized shared variance for MUs in SG, G, and IG layers at the same six stimulus diameters as in the left panel. *Dots*: Shared variance values for individual MUs; *red horizontal lines*: medians. The extent of the box marks the quartiles of the data distribution, and the whiskers mark the width of the entire distribution, save for data points that are considered outliers because they are outside 1.5 times the inter-quartile range. (**C**) Median percent change in shared variance relative to baseline induced by a stimulus matched in size to the RF diameter for the different layers. *Dots here and in (***D–F***)*: Individual data points. Error bars: standard deviation of the bootstrapped distribution of medians. *Black and white dots* indicate data from different animals. (**D**) Median percent change in shared variance relative to a stimulus matched in size to the RF induced by a stimulus matched in size to twice the RF diameter (i.e. near surround) for the different layers. Other conventions as in panel C. (**E**) Median percent change in shared variance induced by a stimulus matched in size to the *per-neuron* surround diameter, relative to shared variance induced by a stimulus with diameter equal to the RF diameter, for the different layers. (**F**) Median percent change in shared variance induced by a 26° diameter stimulus relative to the shared variance evoked by a stimulus matched to the RF diameter, for different layers. (**G**) Median stimulus diameter at the smallest shared variance value (i.e. at max quenching), normalized to the RF diameter of the recorded MUs, for different layers. Error bars: 68% bootstrapped confidence interval of the median (can be asymmetric but otherwise corresponds to bootstrapped standard error). Other conventions as in panel (**C**). (**H**) Median stimulus diameter at the largest shared variance value in the RF surround (i.e. at max facilitation), normalized to the RF diameter of the recorded MUs, for different layers. By construction of the analysis, the smallest possible value is 1. Conventions as in panel (**G**).

The online version of this article includes the following figure supplement(s) for figure 5:

**Figure supplement 1.** Mean-matched Factor Analysis.

15.5%, p ≥ 0.12). The effect on shared variance of stimulating the far surround with a 26° stimulus differed across layers (*Figure 5F*). In SG and G layers, the percent change in shared variance was not significantly different from zero (*Figure 5F*; median percent change in shared variance ± s.d. of the bootstrapped distribution, SG: 6.35 ± 16.97, G: 7.62 ± 36.43, one-tailed bootstrap test p > 0.18). However, in IG layers, the 26° diameter stimulus significantly decreased shared variance compared to its value for a stimulus matched to the RF (median percent change in shared variance ± s.d. of the bootstrapped distribution –33.7 ± 10.99%, one-tailed bootstrap test p < 0.001; *Figure 5F*).

The median stimulus diameter at the lowest shared variance (i.e. at maximum shared variance quenching) normalized to the RF diameter was 1.86 (1.30, 14.7, 99% CI) in SG layers, 2.22 (1.40, 14.9) in G layers, and 6.59 (2.06, 19.9) in IG layers (*Figure 5G*).

The median surround stimulus diameter evoking the highest shared variance, normalized to the RF diameter, was 5.8 (1.0, 24.4) in SG, 9.4 (2.27, 45.0) in G, and 1.49 (1.0, 6.60) in IG (*Figure 5H*).

To better understand the connection between shared variance and changes in firing rate related to stimulus size, we computed the average shared variance following the same mean-matching procedure performed for Fano-factor (see Methods). This analysis showed similar results to those of the non-mean-matched analysis (see Supplementary Methods, Supplementary Results, and *Figure 5— figure supplement 1*).

## Discussion

Using laminar multi-electrode array recordings, we have studied how single neuron response variability and the shared variability among neurons are modulated by stimulus size across the layers of macaque V1. We found that both forms of variability are size-tuned, and this tuning is layer dependent. In all layers, variability declined as a stimulus was progressively increased in size from 0.1° to the diameter of the RF. However, as the stimulus was enlarged beyond the RF, variability changed in a layer and stimulus-size-dependent manner. Stimuli involving the near surround decreased variability and shared variability relative to their values for a stimulus the size of the RF, mainly in the IG layers. In contrast, larger stimuli extending into the far surround increased both single neuron and shared

variability in SG layers (relative to their value for a stimulus matched to the RF diameter), but did not change them or reduced them in G and IG layers. Given the hypothesized influence of variability on visual information processing and the encoding of sensory inputs, these laminar differences suggest that the different layers employ different strategies for coding large stimuli. Moreover, given known laminar differences in connectivity, these laminar-specific effects of stimulus size on variability suggest different underlying circuit mechanisms.

Theoretical work has shown that correlated variability can be detrimental for sensory processing (*Abbott and Dayan, 1999*; *Averbeck et al., 2006*). Consistent with this idea, a number of top–down modulations thought to improve sensory processing and perception tend to reduce spike-count variability (Fano-factor) and correlated variability. For example, attention directed toward a visual stimulus reduces Fano-factor and correlated variability in primate areas V4 (*Cohen and Maunsell, 2009*; *Mitchell et al., 2009*) and V1 (*Herrero et al., 2013*), decorrelation is induced by perceptual learning in area MSTd (*Gu et al., 2011*), and surround suppression reduces Fano-factor (*Festa et al., 2021*) and correlated variability (*Snyder et al., 2014*) in V1. However, theoretical and experimental work has also indicated that not all correlations impede encoding (*Averbeck et al., 2006*), that the strength of variability and correlations depends on stimulus, cognitive factors, cortical layers, and area (*Hansen et al., 2012*; *Smith et al., 2013*; *Ruff and Cohen, 2016a*; *Ruff and Cohen, 2016b*), and that some form of correlations can facilitate, rather than impede, perception (*Ruff and Cohen, 2014*; *Haefner et al., 2016*). In line with these later studies, we find that the effects of surround suppression on variability and correlated variability depend on stimulus size and cortical layer. Our results in IG layers that near and far-surround stimulation reduce Fano-factor and correlated variability are consistent with previous reports (*Snyder et al., 2014*; *Orbán et al., 2016*; *Festa et al., 2021*) (although we used factor analysis as a measure of shared variability, unlike Snyder et al., which instead measured correlations). However, we have additionally shown that the effect of surround suppression on Fano-factor and shared variability depends on both the size of the surround stimulus (near vs. far) and cortical layer. Unlike in IG layers, in G and SG layers, near-surround stimulation did not decrease Fano-factor or shared variability below their values measured for a stimulus matched to the RF size, while far-surround stimulation increased variability in SG layers, but decreased it (or had no effect) in IG layers. As near- and far-surround suppression are thought to be mediated by different circuits (intra-areal horizontal and inter-areal feedback connections, respectively), and different layers have distinct connectivity, these results suggest that different circuits affect variability in different ways. In particular, they suggest that increased activity of long-range intra-areal or inter-areal feedback connectivity in IG layers quenches variability. In contrast, corticocortical feedback connections to the SG layers may be a source of variability and correlated variability in these layers, and enhancing their activity amplifies variability and correlations. It is unlikely that the increased variability in SG layers induced by presentation of large stimuli is detrimental for visual processing. The impact of surround modulation on visual processing ultimately depends on the way neuronal responses are read out, which may vary with cortical layer. It is also possible that variability increases in SG layers and decreases in IG layers induced by surround stimulation both facilitate encoding and perception. Additional studies are necessary to determine how these laminar-specific modulations of variability affect perception and behavioral performance, but these laminar differences are consistent with mounting evidence that SG and IG layers represent functionally distinct processing streams. For example, SG and IG layers show distinct dynamics and spectral behavior (*Maier et al., 2010*; *Wang, 2010*; *Buffalo et al., 2011*; *Mendoza-Halliday et al., 2024*), which are thought to underlie functional and computational asymmetries (*Bastos et al., 2012*; *Shipp, 2016*).

It has recently been suggested that variability and response suppression share a common origin, as manipulations that increase response suppression typically quench variability (*Goris et al., 2024*). Our results, however, indicate that surround suppression is not always associated with variability quenching, as far-surround stimulation in the SG layers increased both variability and shared variability, and instead suggest multiple circuits and mechanisms as the source of variability.

It is often assumed that the trial-to-trial variability of neural responses follows Poisson statistics (*Simoncelli et al., 2004*). Under Poisson statistics, variability does not depend on firing rate; rather, the Fano-factor remains constant (=1) regardless of mean firing rate. In contrast, we found that, across all layers, firing rate increased, and Fano-factor decreased as the stimulus diameter was increased from 0.1° to a diameter equal to that of the RF. These results are inconsistent with the Poisson model

of neural response statistics. An extension of the Poisson model, the modulated Poisson model (**Goris et al., 2014**), augments the Poisson model by a stochastic gain variable. With this addition, the model captures overdispersion (Fano-factor >1) of neural responses in a physiologically and statistically meaningful way. The modulated Poisson model predicts that Fano-factor increases as firing rate increases (**Goris et al., 2014**). In contrast to this prediction, our data show that for stimuli within the RF, Fano-factor decreases as mean firing rate increases. Our results for small stimuli resemble those of **Coen-Cagli and Solomon, 2019** who showed that, in macaque V1, Fano-factor decreased as stimulus contrast (and firing rate) increased. These authors also concluded that their data was incompatible with the modulated Poisson model and suggested that neural response statistics in macaque V1 are better captured by stochastic normalization models.

A different class of models is based on the idea of perception as probabilistic inference (**Knill and Pouget, 2004**). In these models, spikes represent samples from probability distributions, and neural response variability reflects the uncertainty of the inferences (**Hoyer and Hyvärinen, 2003**; **Fiser et al., 2010**; **Orbán et al., 2016**; **Echeveste et al., 2020**; **Festa et al., 2021**). The model by **Festa et al., 2021** predicts a decrease in Fano-factor induced by surround stimulation relative to its value for a stimulus matched in size to the RF. These authors' neural recording results confirmed the model predictions for stimuli extending into the near surround (2xRF size), but for larger stimuli involving the far surround, Fano-factor increased back to its value at the RF size. However, these authors did not examine laminar differences, and the use of 1 mm shank length Utah electrode arrays in their study suggests they may have sampled neurons primarily from the G layer. Our mean-matched analysis confirmed the result of Festa et al. that near-surround stimulation reduces Fano-factor, but we found this to occur only in G and IG layers; we additionally showed that in G and IG layers, as the stimulus was extended into the far surround, Fano-factor either increased back to its value at the RF size (non-mean-matched analysis) or remained at its value for a stimulus in the near surround (mean-matched analysis). This points to the intriguing possibility that in G and IG layers, a significant proportion of neurons serves to perform probabilistic inference. **Festa et al., 2021** also presented a version of their model in which global context had an additive, instead of multiplicative, effect on neural responses. Interestingly, the additive version of their model predicted that stimulating the RF surround should increase variability, as we found in many cells, particularly in the SG layers.

The laminar differences in variability found in our study suggest different underlying circuit mechanisms and potentially different computational functions. One plausible hypothesis is that laminar-specific inhibitory circuits underlie the different effects of surround stimulation on variability. This hypothesis is based on the assumption that inhibition plays an important role in modulating cortical response variability, as postulated by several models (**Stringer et al., 2016**; **Hennequin et al., 2018**; **Huang et al., 2019**), and the well-established laminar differences in the distribution of inhibitory neuron types (**Tremblay et al., 2016**). In macaque sensory-motor cortex, somatostatin-positive inhibitory interneurons predominate in SG layers (**Hendry et al., 1984**; **Gielow et al., 2025**). In mouse cortex, these interneuron types subtractively control the gain of their target neurons (**Sturgill and Isaacson, 2015**), by hyperpolarizing the dendrites of pyramidal cells (**Markram et al., 2004**; **Pouille et al., 2013**), and mediate surround suppression (**Adesnik et al., 2012**). This subtractive inhibition counteracts the excitatory feedforward drive (**Pouille et al., 2013**), thus, ultimately affecting neural responses in a manner resembling reduced feedforward input. Because reduced feedforward input to the cortex increases neural response variability (e.g. **Churchland et al., 2010**; **Festa et al., 2021**, this study), activation of somatostatin neurons ultimately would lead to increased neural response variability. This hypothesis predicts that neurons in which variability is increased by surround stimulation, which are present in all layers but dominate in the SG layers, are those in which surround suppression is mediated by somatostatin cells. Alternatively, reduced feedforward drive induced by surround stimulation, leading to increased variability, may result from withdrawal of feedforward excitation from surround-suppressed excitatory neurons in the thalamus or cortex itself, a mechanism that is consistent with a recent model of neural response variability (**Bressloff, 2019**). This hypothesis predicts that neurons in which variability is increased by surround stimulation are those inheriting surround suppression from other suppressed excitatory neurons. In contrast, neurons for which surround stimulation does not affect or decreases variability (more numerous in G and IG layers) may be surround suppressed via different circuit mechanisms; for example, via inhibitory cells that track

the activity of excitatory cells (such as parvalbumin interneurons). This mechanism quenches variability in a stochastic inhibition-stabilized network model (*Hennequin et al., 2018*).

Quenching of cortical response variability by stimulus onset is considered to be a universal property of the cortex (*Churchland et al., 2010*). In line with this idea, we showed that presenting a stimulus most commonly quenched variability compared to pre-stimulus baseline. However, in addition to variability quenching, we found that, for a substantial fraction of neurons in all layers, but predominantly in G and IG layers, small stimuli amplified variability relative to pre-stimulus baseline. The use of very small stimuli and a unit-by-unit analysis were the key differences between our study and previous studies that did not observe amplification of variability by small stimuli. Amplification of cortical response variability by small visual stimuli relative to pre-stimulus baseline was predicted by a supralinear stabilized network model of cortical response variability (*Hennequin et al., 2018*). This model predicts variability amplification when the stimulus-evoked response is of comparable magnitude to spontaneous activity. However, in our data, variability amplification was observed also when neural responses were significantly above the spontaneous baseline. Thus, the prediction of supralinear stabilized network models of cortical response variability may not be in quantitative agreement with the results of this study. Variability amplification can trivially arise in units with close to zero baseline firing rates, as it is often the case for anesthetized primate V1, because the variance of a spike train with zero mean is necessarily zero and can only increase as the firing rate increases. In our dataset, the cells that showed variability amplification had lower baseline firing rates and Fanofactor values, but also showed variability quenching for larger stimuli. Thus, a floor effect due to low baseline firing rates cannot explain the variability amplification in our data. A potential caveat of our dataset is that even though the animals were anesthetized and paralyzed, small eye movements can still occur, causing a small stimulus to move in and out of the RF. This could cause amplification of variability. While we cannot rule out this explanation for all penetrations, we found amplifying units in penetrations in which eye movements could be confidently ruled out. Thus, we conclude that small, weak stimuli can amplify trial-to-trial variability relative to baseline.

A number of different dynamical models have been proposed to explain various aspects of stimulus-dependent variability. In models with multi-stable dynamics, response variability arises from the stochastic wandering across the cortex of spontaneously formed tuning curves or bumps (*Ponce-Alvarez et al., 2013*; *Bressloff, 2019*; *Huang et al., 2019*). In these models, increased stimulus drive reduces wandering of the activity patterns, locking the stimulus-driven bump in place, and as a consequence, quenching variability. Recently, one such model explicitly predicted an increase in variability by surround stimulation (*Bressloff, 2019*). This prediction is consistent with our results in SG layers, but not in G and IG layers, although this model captures well the experimentally observed differences in the magnitude of variability across cortical layers in the spontaneous state (*Smith et al., 2013*). In a different class of models, instead, variability results from fluctuations about a single, stimulus-driven attractor in a stochastic stabilized supralinear network (*Hennequin et al., 2018*). In these models, when stimulus drive increases, the balanced network causes an increase in inhibition which leads to reduced variability. Thus, in these models, variability quenching results from increased inhibition, as opposed to the multiple-attractor models described above in which variability quenching results from increased excitation. Although the effects of surround suppression on variability have not been explicitly studied in stabilized supralinear network models, they would seem consistent with our results in G and IG layers, that is a reduction or saturation of variability by surround stimulation, but not in SG layers.

In summary, existing models of cortical response variability are either inconsistent or only partly consistent with our results, capturing the effects on variability of stimulus size we have observed for some, but not all, layers. We suspect that both kinds of mechanisms may occur, depending on particular cortical operating conditions and the specific layer. Therefore, our results call for the extension of these existing models or the development of new models that can capture the laminar differences in the stimulus-dependent modulation of cortical response variability we have observed in our study.

# Materials and methods

## Experimental model

Linear array recordings were made in the parafoveal representation (4–8° eccentricity) of V1 in two anesthetized adult macaque monkeys (*Macaca fascicularis, 1 male, 1 female*, 3–4 kg). Here we report recordings from a total of 82 visually responsive MUs (82 out of the 120 recorded channels contained a visually responsive MU) from 5 array penetrations. Two of the penetrations were recorded in one animal, and three in the other animal. For the SU analysis, SUs were spike sorted (*n* = 60 SUs; see Supplementary Methods). The data reported in this paper is a subset of the data previously reported in *Bijanzadeh et al., 2018*. Here, we re-analyzed data from penetrations in which we had collected 21–51 trials per stimulus condition. All experimental procedures were in accordance with Protocol No. 2112014 approved by the University of Utah Institutional Animal Care and Use Committee and with NIH guidelines.

## Surgery

The surgical procedures are described in detail in our previous study (*Bijanzadeh et al., 2018*). Briefly, anesthesia was induced with ketamine (10 mg/kg, i.m.). An intravenous catheter and endotracheal tube were inserted, the head fixed in a stereotaxic apparatus, and the animal was artificially ventilated with a 70:30 mixture of $O_2$ and $N_2O$. End-tidal $CO_2$, blood $O_2$ saturation, electrocardiogram, blood pressure, lung pressure, and body temperature were monitored continuously. A small craniotomy and durotomy were performed over the opercular region of V1, and a PVC chamber was glued to the skull surrounding the craniotomy and filled with agar and silicon oil to prevent cortical pulsation and dehydration, respectively. On completion of the surgery, and after a stable plane of anesthesia was reached, the animal was paralyzed with vecuronium bromide (0.3 mg/kg/h, i.v.), to prevent eye movements. Recordings were performed under continuous infusion of sufentanil citrate anesthesia (4–12 µg/kg/h). The pupils were dilated with topical atropine, and the corneas were protected with gas-permeable contact lenses. The eyes were refracted using corrective lenses, and the foveae were plotted on a tangent screen using a reverse ophthalmoscope and periodically remapped throughout the experiment.

## Electrophysiological recordings

To record the activity of V1 SUs and MUs across cortical layers, 24-channel linear arrays (V-Probe, Plexon, Dallas, Texas, 100 µm contact spacing and 20 µm contact diameter) were inserted into area V1, perpendicular to the pial surface to a depth of 2.0–2.2 mm. A custom-made guide tube provided mechanical stability to the array. To facilitate post-mortem visualization of the lesion tracks, the probes were coated with DiI (Molecular Probes, Eugene, OR) prior to insertion. We recorded extracellularly MU spiking activity and local field potential (LFP). The signals were amplified, digitized, and sampled at 30 kHz using a 128-system (Cerebus, 16-bit A-D, Blackrock Microsystems, Salt Lake City, UT).

## MU selection

MU activity was detected by band-pass filtering continuous voltage traces and thresholding the filtered trace at four times the background noise standard deviation, estimated as the median of the continuous recording divided by 0.6745 (*Quiroga et al., 2004*). The analyses were done only on MUs in which the most strongly driving stimulus evoked at least three spikes above the spontaneous activity (count window 50–350 ms after the stimulus onset). Moreover, only MUs in which the response was tuned for stimulus size were analyzed. Whether a unit was statistically significantly tuned for stimulus size was determined by performing ANOVA on the stimulus-evoked spike counts. The MUs in which the effect of stimulus size was statistically significant (one-way ANOVA, $p < 0.05$) were considered size-tuned.

## Visual stimuli

Visual stimuli were generated using Matlab (Mathworks Inc, Natick, MA; RRID:SCR_001622) and presented on a calibrated CRT monitor (Sony, GDM-C520K, 600 × 800 pixels, 100 Hz frame rate, mean luminance 45.7 cd/m², at 57 cm viewing distance), and their timing was controlled using the

ViSaGe system (Cambridge Research Systems, Cambridge, UK; RRID:SCR_000749). All stimuli were displayed for 500 ms, followed by 750 ms interstimulus interval.

We quantitatively mapped the mRF of MUs across contacts by flashing a 0.5° black square stimulus over a 3 × 3° visual field area. The aggregate mRF of the column was defined as the visual field region in which the square stimulus evoked a mean response (+2 s.d. of the stimulus-evoked response) that was >2 s.d. above mean spontaneous activity, and the geometric center of this region was taken as the MUs' aggregate mRF center. All subsequent stimuli were centered on this aggregate mRF. We then determined orientation, eye dominance, spatial and temporal frequency preferences of cells across contacts using 1–1.5° diameter drifting sinusoidal grating patches of 100% contrast presented monocularly. Subsequent stimuli were presented at the optimal parameters for most MUs across the column. We measured size tuning across the column using 100% contrast drifting grating patches of increasing size (0.1–26°) centered over the aggregate mRF of the column. To monitor eye movements, the mRFs were remapped by hand approximately every 10–20 min and stimuli re-centered on the mRF if necessary. To ensure that the array was positioned orthogonal to the cortical surface, we used as criteria the vertical alignment of the mapped mRFs at each contact, and the similarity in the orientation tuning curves across contacts. If mRFs were misaligned across contacts, the array was retracted and repositioned.

## Quantification and statistical analysis

### CSD analysis

We used CSD responses to small stimuli flashed inside the RFs to identify laminar borders (as detailed in the Results). CSD analysis was applied to the band-pass filtered (1–100 Hz) and trial-averaged LFP using the kernel CSD toolbox (kCSD_Matlab) (*Potworowski et al., 2012*). CSD was calculated as the second spatial derivative of the LFP signal. To estimate CSD across layers, we interpolated the CSD every 10 µm. The CSD was baseline corrected (Z-scored). In particular, we normalized the CSD of each profile to the s.d. of the baseline (defined as 200 ms prior to stimulus onset) after subtraction of the baseline mean (see *Bijanzadeh et al., 2018* for details).

### Fano-factor

To quantify trial-to-trial variability, we computed Fano-factor by dividing the spike-count variance by the mean spike-count over trials. A small constant (0.0000001) was added to the mean spike count to avoid dividing by zero. During the course of developing the analysis, we also used a method in which spike-count variance was plotted against mean spike-count computed over trials in 100 ms non-overlapping bins, and by fitting to the variance-to-mean curves a line so that the intersection of the line and the y-axis was constrained to be zero and the slope of the line was taken as the Fano-factor. All findings of the study were replicated using both methods, but we chose the direct division for convenience, as it allows for more efficient bootstrapping of errors. All of our analyses were performed between 50 and 450 ms after stimulus onset, except for the pre-stimulus baseline that was computed from –400 to 0 ms before stimulus onset.

To determine the significance of the different effects of surround stimulation on Fano-factor (the data in *Figure 2D–F*), we re-sampled % change in Fano-factors 9999 times with replacement from a distribution with zero median, and computed the proportion of samples in which the % change was larger than the observed values.

To determine whether a stimulus caused a statistically significant increase or decrease in Fano-factor relative to baseline in MUs (the analyses presented in *Figure 3*), the distribution of the difference between Fano-factor and baseline at each stimulus size was resampled with replacement 3000 times. The mean of this distribution was set to zero. If the Fano-factor measured at a given stimulus diameter was higher (smaller) than the 95th (5th) percentile of this distribution, we concluded that the stimulus significantly increased (decreased) Fano-factor relative to baseline.

For details on the mean-matched Fano-factor analysis, see Supplementary Methods.

### Function fitting, RF and surround size, and suppression index

To estimate the size of the RF and surround for each unit, we measured size tuning as described above and plotted the mean firing rate of the unit against stimulus diameter; we then fitted these data with ratio-of-Gaussians functions (*Cavanaugh et al., 2002*). The Fano-factor data was fitted with two

ratio-of-Gaussians functions that were summed. These two ratio-of-Gaussians functions had independent parameters. The parameters were optimized by minimizing the squared difference between the function and the data. For firing rate, the minimization was performed with the Levenberg–Marquardt algorithm as implemented in SciPy (*Virtanen et al., 2020*). The parameters of the function were constrained to be positive, including zero. For the Fano-factor, the parameters of the function were fitted with the basinhopping algorithm as implemented in SciPy. As the sum of two ratio-of-Gaussians functions was overfitting the data, we constrained the parameters to be always positive with an upper bound between 1 and 100, depending on the parameter. With these constraints, the fitted functions were always smooth. Two ratio-of-Gaussians functions were also fitted to the shared variance data.

From the fitted functions, the size of the RF was taken to be the stimulus diameter at which the function peaked. The 'per-neuron' surround size was taken to be the smallest stimulus diameter, larger than the RF size, at which the slope of the fitted size-tuning function was at least 10% higher than the minimum slope of the curve. The slope was computed at all stimulus sizes between the RF size and 26°. We used this 'unconventional' measure of surround size because, compared to conventional measures, it seemed to better match qualitative estimates of surround size obtained by manual inspection of the data. Surround sizes obtained with this method were, however, very similar to those obtained by conventional measures.

The suppression index, used to quantify the strength of surround suppression in *Figure 2I–K* was defined as SI = $(R_{RF} − R_{SUR})/R_{RF}$, where $R_{RF}$ is the response at the RF size and $R_{SUR}$ is the response at a grating diameter of 26°, extracted from the fitted functions. The SI equals zero when there is no surround suppression and 1 when the response is completely suppressed.

## Factor Analysis

We used Factor Analysis to decompose the trial-to-trial spike-count covariance matrix into private (single neuron spiking variability) and shared (network) components. Factor analysis was separately performed for each penetration, stimulus condition, and layer. A 300-ms window was used. The covariance matrices were modeled as the product of the factor loading matrix and its transpose, plus a diagonal matrix containing the variances that are private to each unit. The matrix of factor loadings and the diagonal private variance matrix were estimated with the Gaussian-process factor analysis toolbox of *Yu et al., 2009*. As an estimate of the shared variance for each unit, we used the diagonal components of the matrix that results from multiplying the factor loading matrix with its own transpose.

To choose the latent space dimensionality of the model, we performed fivefold cross-validation separately for each stimulus condition. For each cross-validation fold, we performed a leave-neuron-out prediction for all neurons and computed the root-mean-square error (RMSE) of the prediction with respect to the recorded spike counts (*Yu et al., 2009*). The RMSE was then averaged over cross-validation folds. In SG layers, at most stimulus diameters, the RMSE decreased as the latent space dimensionality increased from 1 to 2. However, the decrease was not statistically significant (*t*-test, p > 0.05). In G layers, the RMSE vs. latent space dimensionality curves were essentially flat. In IG layers at all stimulus diameters, RMSE monotonically increased as the latent space dimensionality was increased. Thus, as there was not sufficient evidence in favor of more than one dimensional latent space, we chose to model our data with one latent space dimension.

## Acknowledgements

We thank Kesi Sainsbury for technical assistance, and Drs. Frederick Federer and Sam Merlin for help with experiments. Supported by grants from the National Institutes of Health to AA (R01 EY026812, R01 EY019743, R01 EY031959, BRAIN U01 NS099702) and to LN (K99/R00 EY029374), the National Science Foundation (IOS 1755431) and the Mary Boesche endowed Chair, to AA, a Postdoctoral Fellowship to LN from the Ella and Georg Ehrnrooth Foundation, an unrestricted grant from Research to Prevent Blindness, Inc and a core grant from the National Institutes of Health (EY014800) to the Department of Ophthalmology, University of Utah.

## Additional information

### Funding

| Funder | Grant reference number | Author |
|---|---|---|
| National Eye Institute | R01 EY026812 | Alessandra Angelucci |
| National Eye Institute | R01 EY019743 | Alessandra Angelucci |
| National Eye Institute | R01 EY031959 | Alessandra Angelucci |
| National Institute of Neurological Disorders and Stroke | U01 NS099702 | Alessandra Angelucci |
| U.S. National Science Foundation | IOS 1755431 | Alessandra Angelucci |
| Mary Boesche Endowment | | Alessandra Angelucci |
| National Eye Institute | K99/R00 EY029374 | Lauri Nurminen |
| Ella and Georg Ehrnrooth Foundation | | Lauri Nurminen |

The funders had no role in study design, data collection, and interpretation, or the decision to submit the work for publication.

### Author contributions

Lauri Nurminen, Conceptualization, Data curation, Software, Formal analysis, Validation, Investigation, Visualization, Methodology, Writing – original draft, Writing – review and editing; Maryam Bijanzadeh, Data curation, Software, Validation, Investigation, Methodology; Alessandra Angelucci, Conceptualization, Resources, Data curation, Software, Supervision, Funding acquisition, Investigation, Visualization, Methodology, Project administration, Writing – review and editing

### Author ORCIDs

Alessandra Angelucci (iD) https://orcid.org/0000-0002-1957-2231

### Ethics

All experimental procedures were in accordance with protocol #21-12014 approved by the University of Utah Institutional Animal Care and Use Committee (IACUC) and with NIH guidelines.

### Decision letter and Author response

Decision letter https://doi.org/10.7554/eLife.86334.sa1
Author response https://doi.org/10.7554/eLife.86334.sa2

## Additional files

### Supplementary files

MDAR checklist

### Data availability

Upon acceptance of the study, the data will be available without restrictions at https://doi.org/10.5061/dryad.hqbzkh1x4. The codes used for this study are available at https://github.com/nurminenlab/variability-sizetuning-analysis, copy archived at *nurminenlab, 2026*.

The following dataset was generated:

| Author(s) | Year | Dataset title | Dataset URL | Database and Identifier |
|---|---|---|---|---|
| Nurminen L, Bijanzadeh M, Angelucci A | 2026 | Data From: Size tuning of neural response variability in laminar circuits of macaque primary visual cortex | https://doi.org/10.5061/dryad.hqbzkh1x4 | Dryad Digital Repository, 10.5061/dryad.hqbzkh1x4 |

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

## Appendix 1

### Supplementary methods

#### Mean-matched Fano-factor analysis

To ensure that the stimulus size-dependent modulation of variability was not a simple consequence of firing-rate modulation, we performed a mean-matched analysis (*Mitchell et al., 2009*; *Churchland et al., 2010*). First, for each time step of the analysis, the firing rates of the recorded MUs were binned separately for the conditions being compared. Second, neurons were randomly removed from the bin containing the larger number of MUs, until the bin had an equal number of MUs for both stimulus conditions. This procedure was repeated for all time steps and firing-rate bins until the number of MUs in each firing-rate bin was equal across the conditions to be mean matched. Finally, Fano-factors for the remaining MUs were computed as described for the main analysis. To compute bootstrapped errors and statistics, this procedure was repeated 9999 times. For statistics, the resulting mean-matched PSTHs were averaged over time before computing the relevant statistics.

#### Single-unit analysis

To analyze the responses of spike-sorted SUs, the voltage traces (sampled at 30 kHz) of each channel were first high-pass filtered at 300 Hz and then spikes were detected and clustered using Kilosort (*Pachitariu et al., 2016*). The clusters were manually refined using the graphical user interface Phy. We accepted units with signal-to-noise ratio (SNR) above 2.5 at the contact with the maximum spike amplitude. SNR is commonly used as a SU quality measure for electrode arrays recordings (*Guillory and Normann, 1999*), and similar SNR thresholds have been used in previous studies on cortical response variability (*Rosenbaum et al., 2017*).

#### Mean-matched Factor Analysis

To mean-match the shared variability data, we first estimated shared variability for all neurons as described in the Results and Methods. Next, mean-matching of firing rates was performed in the same way as described above for the mean-matching Fano-factor analysis. Finally, for the units that survived mean-matching, shared variability was averaged and plotted as shown in *Figure 5—figure supplement 1*.

### Supplementary results

#### Mean-matched Fano-factor analysis

In all layers, the mean-matched Fano-factor analysis showed a statistically significant decrease in Fano-factor for a stimulus diameter matched to the RF size, evoking a higher mean spike count, relative to a 0.1–0.2° diameter stimulus (mean ±bootstrapped 99% CI of the mean-matched Fano-factor PSTH at 0.1–0.2° vs. RF stimulus diameter: SG, 2.41 ± 0.06 vs. 1.83 ± 0.05, p < 0.0001; G, 1.58 ± 0.05 vs. 1.39 ± 0.06, p < 0.0001; IG, 1.90 ± 0.04 vs. 1.75 ± 0.04, p < 0.0001; first and fifth columns in *Figure 2—figure supplement 2A–C*).

For the SG layers, the mean-matched Fano-factor showed a significant increase, relative to a stimulus matched to the RF diameter, for all stimuli extending into the RF surround (*Figure 2—figure supplement 2A*; mean ± bootstrapped 99% CI of mean-matched Fano-factor at RF vs. 2xRF: 1.87 ± (0.05, 0.06) vs. 1.95 ± (0.06, 0.06); RF vs. 4xRF: 1.94 ± (0.07, 0.07) vs. 2.36 ± (0.07, 0.06); RF vs. 26° stimulus 1.65 ± (0.06, 0.09); p < 0.0001, bootstrap test for the difference in mean, with the null hypothesis that the samples come from the same distribution).

The converse was true for G and IG layers, where the Fano-factor showed a significant decrease, relative to a stimulus matched to the RF diameter, for stimuli extending into the RF surround (*Figure 2—figure supplement 2B, C*; G: RF vs. 2xRF 1.06 ± (0.04, 0.05) vs. 1.01 ± (0.03, 0.03), p = 0.003, RF vs. 4xRF 1.17 ± (0.05, 0.08) vs. 1.17 ± (0.03, 0.04), p = 0.261, and RF vs. 26° diameter 1.35 ± (0.09, 0.11) vs. 1.16 ± (0.05, 0.05), p = 0.008; IG: RF vs. 2xRF 1.47 ± (0.03, 0.04) vs. 1.27 ± (0.03, 0.03), p < 0.0001, RF vs. 4xRF 1.50 ± (0.03, 0.03) vs. 1.23 ± (0.02, 0.03), p < 0.0001, and RF vs. 26° stimulus diameter 1.55 ± (0.03, 0.04) vs. 1.19 ± (0.03, 0.03), p < 0.0001).

## Single-unit analysis

We replicated the main analyses of this study for spike-sorted SUs. Our SU sample was smaller than our MU sample and consisted of 35 SG, 24 IG, and 1G layer SUs. As we had just one G layer unit, we did not perform this analysis on G layer units, as meaningful statistical analyses could not be performed. Moreover, we removed from this analysis 7 SG and 8 IG SUs whose Fano-factor tuning curve was noisy and essentially flat. Thus, the SU analysis was performed on a total of 28 SG and 17 IG SUs.

Individual SUs showed a similar variety of size tuning curves as the MUs (*Figure 2—figure supplement 3A*). Similar to the MUs, for the majority (75% for SG and 59% for IG) of SUs, Fano-factor decreased as the stimulus diameter was increased to fill the RF (e.g. SG units # 44, 9, and IG units # 5, and 82).

Similar to the MUs, surround stimulation affected Fano-factor in three distinct ways: in all layers, in response to near or far-surround stimulation, individual SUs could show increases, decreases, or no change in Fano-factor relative to its value for a stimulus matched to the RF, but the proportion of SUs showing each type of behavior varied somewhat between layers. To quantify this observation, similar to the MU analysis in *Figure 2B*, we fitted functions to the individual SU Fano-factor size tuning curves and extracted from the fitted curves the Fano-factor measured at different stimulus sizes (*Figure 2—figure supplement 3B*). In both SG and IG layers, the Fano-factor slightly decreased as the stimulus diameter was increased to 2x RF diameter (mean normalized Fano-factor ± bootstrap s.e., *t*-test on log-transformed data, RF vs. 2RF SG: $1.0 \pm 0$ vs. $0.93 \pm 0.03$, p = 0.03, IG: $1.0 \pm 0$ vs. $0.91 \pm 0.06$, p = 0.009). For larger stimuli, involving the far surround, in the IG layers, the Fano-factor plateaued to its value for near-surround stimuli, whereas in the SG layers, it increased back to larger values at the largest stimulus diameter (26°).

In both SG and IG layers, the population median stimulus diameter that evoked the lowest Fano-factor value (max quenching) in SUs was approximately twice the size of the RF (the near surround; *Figure 2—figure supplement 3C*). The median (±bootstrapped s.e) stimulus diameter at lowest Fano-factor value normalized to the RF diameter was $2.5 \pm 1.7$ in SG, and $3.0 \pm 2.1$ in IG layers. However, there was considerable unit-to-unit variability in the stimulus diameter evoking the lowest Fano-factor value, as demonstrated by the individual data points in *Figure 2—figure supplement 3C*; while the lowest Fano-factor value was often measured for a stimulus that extended into the RF surround, in a considerable proportion of SUs, the smallest Fano-factor was measured for a stimulus equal to, or smaller than, the RF. *Figure 2—figure supplement 3D* shows the normalized stimulus diameter that evoked the highest Fano-factor. For this analysis, stimulus diameters equal to or larger than the RF were included. In SG layers, the median (±bootstrapped s.e.) normalized stimulus diameter at the highest Fano-factor was 3.2 (1, 6.6; 68% bootstrapped confidence interval, p = 0.04), while in IG layers, it was 1.8, 2.6, 5.3; p = 0.22. Thus, similar to the MU data, in SG layers, far-surround stimulation increased Fano-factor relative to Fano-factor for a stimulus in the RF, whereas it had no effect in IG layers.

## Mean-matched factor analysis

Mean-matched shared variance analysis produced similar results to the non-mean-matched analysis (*Figure 5—figure supplement 1*). In all layers, the shared component of variability decreased when the stimulus diameter was increased from 0.1° to a size matching the RF diameter (0.1° vs. RF mean ± s.e.m., SG: $0.73 \pm 0.22$ vs. $0.25 \pm 0.09$, G: $0.79 \pm 0.17$ vs. $0.15 \pm 0.03$, IG: $0.53 \pm 0.17$ vs. $0.25 \pm 0.05$, p < 0.03). Near-surround stimulation (2xRF) had no significant effect on shared variance in any layer (p > 0.06). Stimulating the far surround had opposite effects on shared variability in SG and G layers compared to IG layers. In SG and G layers, increasing the stimulus diameter beyond the RF increased shared variance. In both layers, the shared variance was significantly higher when the stimulus diameter was 26° (but not when it was 4xRF) compared to when it matched the RF diameter (RF vs. 4xRF mean ± s.e.m, SG: $0.20 \pm 0.05$ vs. $0.37 \pm 0.16$, p = 0.09; G: $0.11 \pm 0.02$ vs. $0.17 \pm 0.03$, p = 0.10; RF vs. 26° mean ± s.e.m. SG: $0.13 \pm 0.04$ vs. $0.61 \pm 0.21$, G: $0.09 \pm 0.02$ vs. $0.19 \pm 0.04$, p < 0.03). In contrast, far-surround stimulation did not affect the shared variance in IG layers (RF vs. 4xRF mean ± s.e.m.: $0.22 \pm 0.04$ vs. $0.16 \pm 0.02$, p = 0.09; RF vs. 26° mean ± s.e.m. $0.23 \pm 0.05$ vs. $0.16 \pm 0.03$, p = 0.15).

