## [Editor Report]

This valuable study examined relationships between stimulus size and response variability in the primary visual cortex of macaque monkeys. The authors present convincing evidence that, contrary to some previous reports, increases in stimulus size that induce surround suppression are not always accompanied by reductions in response variability across layers of visual cortex. These layer-specific patterns of response variability will help inform and constrain our understanding of functional circuitry in visual cortex.

---

## [Decision Letter]

**Decision letter after peer review:**

[Editors’ note: the authors submitted for reconsideration following the decision after peer review. What follows is the decision letter after the first round of review.]

Thank you very much for submitting the paper "Size tuning of neural response variability in laminar circuits of macaque primary visual cortex" for consideration by *eLife*. Your article has been reviewed by 2 peer reviewers, and the evaluation has been overseen by a Reviewing Editor and a Senior Editor. The reviewers have opted to remain anonymous.

Comments to the Authors:

We are sorry to say that, after consultation with the reviewers, we have decided that this work will not be considered further for publication by *eLife*.

Both reviewers identified significant methodological and theoretical concerns. It was discussed whether these concerns could potentially be addressed, but they are substantive and would change this into a new study that would need to be assessed on its own merits rather than constitute a revision.

*Reviewer #1 (Recommendations for the authors):*

This manuscript presents an experimental characterization of response variability in the macaque primary visual cortex, focused on two factors that affect variability, namely the laminar location of the neuron and the size of visual stimuli.

The authors use laminar probes to record simultaneously from tens of neurons across different V1 layers. After mapping the receptive fields (RF), they measure size tuning: they present gratings centered on the aggregate RF of all neurons recorded, and they vary the size of the stimuli from much smaller to much larger than the RF. They report that the mean firing rate is modulated by stimulus size similarly across layers. They also report that response variability is modulated by stimulus size, but the modulation depends on the cortical layer. They discuss how these findings relate to prior studies of contextual modulation of variability, and suggest that existing circuit models could explain the modulation observed in some layers but not others.

This manuscript is of interest to a specialized audience, but it does not seem appealing to a broader neuroscience audience. The interpretation of the findings appears somewhat simplistic and could be expanded to reflect past knowledge of surround mechanisms, including by the authors themselves. Additional analyses could strengthen the conclusions.

Summary of strengths:

Understanding the factors that affect response variability in the cortex is important because variability plays a central role in many theories of neural coding. Previous studies have identified many factors that modulate variability, but only a few have focused on stimulus size, and those did not investigate layer-specific effects. The findings of this manuscript, that size tuning of variability depends on the cortical layer, are interesting because they could inform future work on the circuit mechanisms that control variability and on the consequences for information coding.

The experiments are well designed and executed, and the dataset will be useful for other researchers interested in size tuning, particularly thanks to the dense sampling of stimulus sizes.

The manuscript is written clearly (but see below for several important concerns).

Summary of weaknesses:

The manuscript is limited to the empirical characterization of a specific phenomenon. Although the authors motivate why layer dependence of size tuning of variability is important, because it challenges mechanistic understanding and questions the implications for stimulus encoding, the work presented here does not propose or test any hypothesis or model that would address those issues. As a result, the contribution of the manuscript is narrow, and it does represent a major conceptual advance in understanding visual information processing in V1.

The manuscript uses the word 'surround' with different meanings and definitions, and the main claims stated are valid only for some of those definitions. I found this confusing, but it could be clarified in the text. In my opinion, bringing out these subtleties would also give space to some interesting results that are in the data but not clearly stated in the text. The authors also focus on a specific stimulus design. This is fine but does not consider many relevant variations in stimulus design that, based on existing literature, could impact the main findings and could be relevant for comparison with prior work. See below for details.

The discussion of some prior related work is inaccurate.

More clarity is needed about 'surround' and RF.

Definitions of surround and RF

The main claim of the manuscript is that surround stimuli increase variability relative to RF-sized stimuli in SG layers, and decrease or do not change variability in G and IG layers. However the word 'surround' refers to different things: the formal definition of the surround diameter for an individual neuron is given in Methods, based on the first stimulus size that leads to a threshold level of firing rate suppression. This definition is consistent with the literature and gives estimates of 2-3 degrees diameter of the suppressive surround. Then the text often refers to surround modulation of variability as the effects of a 26 degree stimulus, which is an order of magnitude bigger than the per-neuron surround. Third, the experiments sample stimulus size very finely, so they afford a fine grained assessment of surround modulation, for many intermediate sizes between the per-neuron RF, per-neuron surround, and 26 degrees.

The analyses and text focus more prominently on the effects of a 26 degree stimulus, but the numbers for maximal quenching (L187, Figure 2F; L337, Figure 5E) indicate that intermediate sizes produce maximal suppression of variability. When one considers the above, the claim that surround stimulation increases variability in SG and decreases/no-change in G/IG is oversimplification. The data suggest a more nuanced picture, where (1) stimulation between the per-neuron RF and per-neuron surround generally reduces variability. (2) stimulation of a much larger visual area generally increase variability (SG) or does not change it (G/IG). This is interesting per se, and also raises the possibility that different mechanisms are engaged by near surround and far surround stimulation, perhaps in line with previous work by the Angelucci lab.

Population plots in Figure 2A are described in terms of population aggregate RF. It would be relevant to present the data also in units of individual RF size, rather than degrees. Although RFs of individual neurons are aligned by design in this experiment, they are not perfectly aligned and presumably they vary in size, so the per-neuron RF and the aggregate RF are conceptually distinct, and the results for each of those have different meanings. As a consequence, analyses of shared variance report effects relative to RF size, but RF in this case can only be the population aggregate. This should be taken into account when interpreting the results.

For completeness, the analyses of Figure 2C,D,G should be repeated also for "surround" stimuli that match the first definition of per-neuron surround (based on the slopes of the size tuning curve).

Figure 2E, caption typo? "stimulation of RF…" should be of the 'surround' instead?

L178 does not specify if 'surround' means 26 degree or per-neuron surround.

L641: The description of surround in Methods L 641 is not clear: isn't the slope at the RF size equal to zero, by definition of RF size as the peak of the size tuning curve?

L41 in the Abstract, I suggest changing 'small visual stimuli' to 'stimuli smaller than the RF'

- Past work on surround modulation of variability. The Discussion text starting on L408 suggests that Festa et al. 2021 report no surround modulation of Fano factor. This is not correct, the main conclusion of that paper is that stimuli extending beyond the RF reduce the Fano factor, consistently with the predicted role of variability for probabilistic inference. Similar surround modulation of variability was also reported by Orban et al. 2016, which is worth discussing in relation to the findings presented here.

- Analysis of shared variance. The analysis of 'shared variance' applies factor analysis separately for each stimulus condition, but always assumes a single latent shared factor. This might not be the correct way to compare shared variance across conditions, because the latent dimensionality could change as well. The choice of dimensionality per condition should be based on cross-validation. The comparison could also be performed dimension per dimension, by comparing the amount of variance explained along each latent dimension.

- Clarity.

Methods inclusion criteria: state the fraction of neurons included ("visually responsive") out of the total channels recorded.

When averaging Fano factors, the geometric mean is more appropriate because FF is a ratio. When averaging changes across conditions, eg. Figure 2C,D, the median seems more appropriate given the outliers.

The error bars in 2F 3C 5E look wrong, not representative of the data distribution.

*Reviewer #2 (Recommendations for the authors):*

Cortical responses to sensory stimuli are variable, and that variability is often shared across a population of neurons. Variability and co-variability are likely to depend on where neurons sit in the cortical hierarchy and the relative weight of feedforward and recurrent connections. Stimulus size offers the opportunity to vary the relative weight of feedforward-excitatory (small stimuli) and recurrent excitatory/inhibitory (large stimuli) inputs to cortical neurons. The authors, therefore, set out to characterise the lamina-organisation of response variability and co-variability, and its dependence on stimulus size, in the primate visual cortex. They find while a stimulus usually suppresses variability and co-variability, both recover slightly at large stimulus sizes, at least for superficial layers of visual cortex.

The measurements are derived from 5 penetrations of a 24-channel linear probe into V1 of 2 anaesthetised macaque monkeys. Multi-unit activity was identified on each channel and subject to analysis. Visual stimuli, usually consisting of simple patterns (gratings) of varying size were presented. The main strength of the work lies in using analyses of data obtained from primates, after careful assignment of recording sites to different layers of the cortex, to address interesting questions about whether processing in the visual cortex depends on which layer neurones are likely to belong to. The main weakness of the work lies in the fact that it relies on multi-unit activity, which is imperfect for the study of neural variability, particularly when variation in the stimulus (such as stimulus size) may result in the addition or removal of units from the MUA. Additionally, because there can be slow drifts in eye-position under anaesthesia, it is difficult to know whether fluctuations in response to small stimuli are due to changes in brain state or eye position.

1. Increase in Fano Factor and shared variability for large stimuli. The authors conclude that variability increases as a stimulus grows larger, particularly in superficial layers. However, many units in superficial layers show complete surround suppression (e.g. Sceniak et al., 2001), and the MUA may be composed of different units when tested with small and large stimuli. This is presumably the case in the current data, as for large stimulus, the evoked rate is apparently no different to the baseline rate on average (Figure 3A), so the stimulus does not recruit an increase in mean activity over baseline, across the population of sites studied, and the Fano factors are derived from units that, on average, are not visually responsive. It, therefore, seems likely that Fano factors for large stimuli are derived from a different (or at least only partially overlapping) pool of neurones to that for smaller stimuli. I am not sure how to address this concern with MUA activity, because of the inherent ambiguity.

2. Amplification of variability for small stimuli. The authors conclude that small stimuli increase variability. If true, this would be of high interest. The concern here is that trial-by-trial variability in mean rate might be confounded with small changes (including slow drifts) in eye-position, which are known to occur even under paralysis unless the eyes are restrained by other means (Forte et al., 2002). This could have different effects on variability as a function of stimulus size, as small but not larger stimuli move into and out of the receptive fields of some or all of the underlying units in the MUA during drift. For example, responses to the 0.1 deg stimulus in Figure 3A may be consistent with drift in the middle third of trials (this data also looks like it is a subset of that in Figure 4A-C, which shows the same pattern of effect for the smallest stimulus). The authors state on line 573: "To monitor eye movements, the RFs were remapped by hand approximately every 10-20 minutes and stimuli re-centered on the RF if necessary." To allow better comparison on such an important point it would be useful to include rasters for the smallest (e.g. 0.2 deg) stimuli giving a clear response in Figure 1. Additionally, given that 0.1 deg stimuli don't seem to elicit a response above baseline (Figure 2A) on average, it would seem better to represent 0.2 deg responses in Figure 2B. In general, however, without a secondary source of evidence that eye-position was stable for the duration of recording, it is difficult to know how to address these findings.

The work could be of interest because it makes measurements from the primate visual cortex and similar measurements have helped understand both vision and more general aspects of cortical function such as lamina-organisation, and the origins and impact of neural variability. Currently, however, the measurements do not provide highly convincing evidence for the conclusions that are drawn.

3. I think these recordings are a subset of those in Bijanzadeh et al., (2018), which is referenced frequently and used similar lamina measurements (from 4 animals) but I am not sure. Please make this explicit. In addition, please clarify how the penetrations from each of the two animals are distributed (e.g. 2 in one animal, 3 in another). Finally, while it might be that statistical tests on the sites from each individual may not be informative, it would be of interest to know whether the effect sizes/directions were the same in each.

4. Spiking activity is defined as multiunit on line 107, but there are several occasions subsequently where words like 'simultaneously recorded units' (line 112), 'example units' (e.g. line 121), 'unit-by-unit' (line 174), 'majority of units' (line 175) are used, where the use of the term unit usually invokes the sense that a single neuron is been discussed. Indeed, on line 303 and in the legend to some Figs, the word 'neuron' is used, or on line 426 '15% of cells' is used. I think 'site' instead of 'unit' or 'neuron' or 'cells' needs to be used throughout, to avoid confusion.

5. The stimulus parameters were chosen to maximise the response of as many sites as possible on the relevant penetration. As there were only 5 sessions it would be good to record those parameters, and perhaps to make sure that the choice did not influence the outcomes (e.g. were particular penetrations associated with particular distributions of variance / co-variance, where that may have been because of the choice of stimulus?).

6. The statement that the stimulus size yielding max quenching differs between layers (lines 188-9) should be accompanied by an appropriate statistical test or moderated.

7. The mean matching conducted in Supplementary Figure seems key, and it deserves inclusion in the main manuscript. To be honest, I have read the figure description several times and am still unclear about what each of the panels shows.

8. I cannot find anywhere the number of trials presented. This is important for understanding the robustness of the variability and particularly co-variability estimates. Must include explicitly, and also around discussion of mean-matching (ie. how many trials are retained in the analysis).

9. It is not clear to me why only sites with at least 5% surround suppression were studied (line 552). This would seem to be likely to bias the results, and perhaps differently in different layers given the generally stronger suppression found in superficial layers. What happens if that filter is removed?

[Editors’ note: further revisions were suggested prior to acceptance, as described below.]

Thank you for resubmitting your work entitled "Size tuning of neural response variability in laminar circuits of macaque primary visual cortex" for further consideration by *eLife*. In the time that has passed since your last submission, the handling editors left our editorial board, so your revised article has been evaluated by Joshua Gold (Senior Editor).

The manuscript has been improved but there are some remaining issues that need to be addressed, as detailed below in the comments from Reviewer 3:

*Reviewer #3 (Recommendations for the authors):*

I appreciate the effort that the authors have put into the reply, and very much understand the delays that normal life imposes. I apologise similarly that it has taken me this long to analyse the responses and revised manuscript. As in my initial assessment, I continue to believe that the main strength of the work lies in using analyses of data obtained from primates, after careful assignment of recording sites to different layers of the cortex, to address questions about layer-dependence of processing in the visual cortex.

The authors have addressed many of my concerns, but while the new analyses of single-unit activity show that variability can depend on size in single-units (including some which show increase in variability for larger sizes) they do not convincingly demonstrate a layer dependence of these effects. Further, even in multi-unit activity, this larger-size-effect appears to be strongest in one or two of the recordings, and weak if present in other recordings. I am therefore not sure that a major conclusion of the abstract ("variability is tuned for stimulus size in a layer-dependent manner.") can be robustly supported by the current data.

In the following I address the authors replies that I remain concerned about:

1. The authors state: "… While the SU data analysis is somewhat less robust than the analysis performed for multi-units (MUs; in Figure 2), due to the smaller sample size, the SU analysis shows similar results as the MU analysis."

Thank you for performing these analyses and including the new data. However, while Figure 2 —figure supplement 3 shows individual units in which variability can increase with stimulus size, on average (panel B of this supplementary data) neither SG nor IG show an increase in Fano Factor at larger sizes. I therefore think that this statement rests on the observation that many of the units in Panel D (which shows relative size at which there is peak facilitation) show values larger than 1 in SG but fewer are larger than 1 in IG. I would appreciate clearer description of this dependency in the main text, and more stringent tests of the hypothesis that in SU as in MU, variability is greater at larger sizes, particularly in SG.

In addition, I believe that reanalyses are required here to ensure that the conclusions are correct -the Supplementary Methods state that 28 SG and 17 IG SUs were included in the analyses (and 7/8 respectively were excluded because the tuning curve for Fano Factor was 'noisy and essentially flat'), and I count 28 SG and 16 IG data points (one may be obscured) in panel B. However I count 31 SG and 24 IG data points in panel C (some may be obscured, and there is too much overlap in panel D to count them). I am therefore concerned that the analyses may have inadvertently included units that should not be included, because their Fano Factors were not adequately tuned to measure the size at the maximum or minimum.

2. The authors state: "Because the RF of the neurons in our sample were precisely aligned retinotopically (see Bijanzadeh et al. 2018 for examples and detailed analysis), a shift in eye-position that could explain the amplification is expected to abolish neural responses across the whole array. To demonstrate that variability amplification can happen independent of eye movements, we show in Figure 3-suppl1 an example penetration that showed variability amplification for a 0.6deg stimulus, with no indication that eye movements may have occurred during that recording."

Thank you for the new figure. I am not sure how to read Figure 3-suppl1, however, as it shows rasters of MU activity for different trials but does not quantify variability among those or the amplification of variability – the figure needs accompanying quantitative analysis to be able to interpret and justify the statement in results that "Figure 3—figure supplement 1 rules out eye movements as a potential of increased variability for stimuli smaller than the RF size". In addition, the influence of small eye movements on trial-by-trial measures of variability will presumably depend on receptive field size, which varies with layer. Influence of eye-movements on variability as a function of time into trial may also depend on phase-sensitivity (i.e. simple-cell like), which can also vary with layer. I don't want to push on this too much, however I do think that the potential influence of small eye-movements/drift should be discussed as a limitation.

3. The authors state: "Where feasible, we have plotted data for the two animals with different symbols (Figure 2 C-H, Figure 2-suppl3C-D, Figure 5C-H), and in Figure 2-suppl1 we show data from each penetration separately."

Thank you for distinguishing the animals in the figures and for including the data for individual recordings in Figure 2 -suppl1. I may be missing something, but my reading of the panels in this supplementary data is that while Fano factor decreases from smallest to larger sizes in most or all penetrations, subsequent increase of Fano factors at larger sizes is strongest in the penetration denoted by red lines, where it is present in all layers but particularly prominent in SG. It is hard to see clear evidence for larger-size-dependence outside this penetration. I therefore think that it is important to perform statistical analyses that take into account the particular penetration on which sites were recorded in assessing layer differences in the increase in Fano Factor at larger sizes. This is likely to be necessary for SU as well as MU analyses.

Other points arising from revisions:

1. Figure 1: Are panels Figure 1B and Figure 1C flipped? The text associated with these on page 6 is more consistent with such an arrangement: "…increasing the stimulus diameter beyond the RF boundaries did not affect Fano-factor for the G layer MU (Figure 1B)", "but for larger stimuli (involving the "far" RF surround), Fano factor increased back to approximately its value for a stimulus matched to the RF size (Figure 1C)".

2. Page 7: "In contrast, in IG layers, the minimum Fano-factor was reached at a stimulus diameter approximately twice the RF diameter (we term this the "near surround"), and beyond this stimulus size, Fano-factor increased back to its value for a stimulus matched to the RF diameter (Figure 2A Right).". It is hard to see this in Figure 2a. Perhaps an arrow to indicate the position of the minimum.

3. Page 7: "In G layers, Fano-factor reached its minimum for a stimulus the size of the RF diameter" While the change from 1x to 2x is only significant in IG, both IG and G reach a minimum at 2xRF.

---

## [Author Response]

[Editors’ note: The authors appealed the original decision. What follows is the authors’ response to the first round of review.]

Reviewer #1 (Recommendations for the authors):This manuscript presents an experimental characterization of response variability in the macaque primary visual cortex, focused on two factors that affect variability, namely the laminar location of the neuron and the size of visual stimuli.The authors use laminar probes to record simultaneously from tens of neurons across different V1 layers. After mapping the receptive fields (RF), they measure size tuning: they present gratings centered on the aggregate RF of all neurons recorded, and they vary the size of the stimuli from much smaller to much larger than the RF. They report that the mean firing rate is modulated by stimulus size similarly across layers. They also report that response variability is modulated by stimulus size, but the modulation depends on the cortical layer. They discuss how these findings relate to prior studies of contextual modulation of variability, and suggest that existing circuit models could explain the modulation observed in some layers but not others.This manuscript is of interest to a specialized audience, but it does not seem appealing to a broader neuroscience audience. The interpretation of the findings appears somewhat simplistic and could be expanded to reflect past knowledge of surround mechanisms, including by the authors themselves. Additional analyses could strengthen the conclusions.

There is widespread interest in variability in the field of neuroscience which encompasses a variety of aspects regarding this phenomenon, from how it is affected by factors such as stimulus type, brain states, cortical layer, cortical area, to its potential function, to the circuit and mechanisms that may generate it and modulate it. Therefore, we believe that our study will be of broad interest to this community. In this revised version, as suggested by this Reviewer, we have re-written the introduction and expanded the discussion on potential circuit mechanisms. We have also performed more extensive analyses.

Summary of strengths:Understanding the factors that affect response variability in the cortex is important because variability plays a central role in many theories of neural coding. Previous studies have identified many factors that modulate variability, but only a few have focused on stimulus size, and those did not investigate layer-specific effects. The findings of this manuscript, that size tuning of variability depends on the cortical layer, are interesting because they could inform future work on the circuit mechanisms that control variability and on the consequences for information coding.The experiments are well designed and executed, and the dataset will be useful for other researchers interested in size tuning, particularly thanks to the dense sampling of stimulus sizes.The manuscript is written clearly (but see below for several important concerns).Summary of weaknesses:The manuscript is limited to the empirical characterization of a specific phenomenon. Although the authors motivate why layer dependence of size tuning of variability is important, because it challenges mechanistic understanding and questions the implications for stimulus encoding, the work presented here does not propose or test any hypothesis or model that would address those issues. As a result, the contribution of the manuscript is narrow, and it does represent a major conceptual advance in understanding visual information processing in V1.

While our study may not provide new insights or suggest specific model/s for how laminar differences affect encoding of visual information, it provides an indirect test of hypotheses and predictions made by several existing models. In particular, our results highlight how most current models fail to explain laminar and size-tuning differences in variability, and suggest that a single mechanism is insufficient to explain these various effects on variability.

The manuscript uses the word 'surround' with different meanings and definitions, and the main claims stated are valid only for some of those definitions. I found this confusing, but it could be clarified in the text. In my opinion, bringing out these subtleties would also give space to some interesting results that are in the data but not clearly stated in the text. The authors also focus on a specific stimulus design. This is fine but does not consider many relevant variations in stimulus design that, based on existing literature, could impact the main findings and could be relevant for comparison with prior work. See below for details.The discussion of some prior related work is inaccurate.More clarity is needed about 'surround' and RF.

We agree with the Reviewer and have expanded the definition of surround, which now includes near and far surround, and performed new analyses at multiple different stimulus sizes. These new analyses indeed provide new insights (see below for a detailed reply). We have also corrected misinterpretations of results from prior studies (see below for a detailed reply).

Definitions of surround and RFThe main claim of the manuscript is that surround stimuli increase variability relative to RF-sized stimuli in SG layers, and decrease or do not change variability in G and IG layers. However the word 'surround' refers to different things: the formal definition of the surround diameter for an individual neuron is given in Methods, based on the first stimulus size that leads to a threshold level of firing rate suppression. This definition is consistent with the literature and gives estimates of 2-3 degrees diameter of the suppressive surround. Then the text often refers to surround modulation of variability as the effects of a 26 degree stimulus, which is an order of magnitude bigger than the per-neuron surround. Third, the experiments sample stimulus size very finely, so they afford a fine grained assessment of surround modulation, for many intermediate sizes between the per-neuron RF, per-neuron surround, and 26 degrees.The analyses and text focus more prominently on the effects of a 26 degree stimulus, but the numbers for maximal quenching (L187, Figure 2F; L337, Figure 5E) indicate that intermediate sizes produce maximal suppression of variability. When one considers the above, the claim that surround stimulation increases variability in SG and decreases/no-change in G/IG is oversimplification. The data suggest a more nuanced picture, where (1) stimulation between the per-neuron RF and per-neuron surround generally reduces variability. (2) stimulation of a much larger visual area generally increase variability (SG) or does not change it (G/IG). This is interesting per se, and also raises the possibility that different mechanisms are engaged by near surround and far surround stimulation, perhaps in line with previous work by the Angelucci lab.

We agree with the Reviewer that the original claim about the effects of surround stimulation on variability was an oversimplification, and thank the Reviewer for suggesting a more fine-grained analysis. We have now performed the analysis at multiple stimulus sizes (RF, 2xRF, 4XRF, 8xRF, per neuron-surround and 26deg); new Figures 2 and 5, and related supplementary Figures 2-suppl2, and Figure 5-suppl1 report the results of this new analysis. In agreement with the Reviewer’s observation, we now find differences in near vs far surround stimulation, and these differences are also layer-dependent. In general, stimuli in the near surround (about 2xRF size) cause a decrease in variability and shared variability although this is more prominent and significant in the IG layers (e.g. Figure 2B-G), and in G layers (in the mean matched analysis -Figure 2-figure suppl). (2). In the new analysis we still find that the far surround causes opposite effects in the SG (increases in variability) vs. IG layers (either no change or decrease). We have corrected all text in the Abstract, Introduction (pp.4-5), Results (pp.7, 8,9,10, 15), and Discussion (pp. 16,17, 18), as well as Supplementary Methods and Results, to reflect this new finding.

It is important to stress, however, that our main conclusion, that the impact of RF surround on variability is different on average in SG compared to G and IG, remains valid.

Population plots in Figure 2A are described in terms of population aggregate RF. It would be relevant to present the data also in units of individual RF size, rather than degrees. Although RFs of individual neurons are aligned by design in this experiment, they are not perfectly aligned and presumably they vary in size, so the per-neuron RF and the aggregate RF are conceptually distinct, and the results for each of those have different meanings. As a consequence, analyses of shared variance report effects relative to RF size, but RF in this case can only be the population aggregate. This should be taken into account when interpreting the results.

Following the Reviewer’s suggestion, we have now performed an analysis in which the stimulus size was normalized to the RF size before averaging (new Figure 2A and new Figure 5A). This shows that in SG layers, the minimum Fano-factor value is found when the stimulus size equals the RF size, and Fano-factor increases as the stimulus encroaches into the RF surround. Instead, in G and IG layers, the population averaged Fano-factor reached a minimum at approximately 2 times the RF size (Figure 2G). After reaching this minimum, in G layers, the Fano-factor first increases and then plateaus, whereas in IG layers the Fano-factor does not change systematically and overall appears to remain at the same Fano-factor value as for the near surround.

For completeness, the analyses of Figure 2C,D,G should be repeated also for "surround" stimuli that match the first definition of per-neuron surround (based on the slopes of the size tuning curve).

As suggested by the Reviewer, we now show in Figure 2D-F the effect on Fano-factor of near-surround (2xRF), per-neuron surround and a 26° stimulus diameter. A similar analysis is shown for shared variance in Figure 5D-F.

Figure 2E, caption typo? "stimulation of RF…" should be of the 'surround' instead?

Thank you. We have now corrected the figure legend.

L178 does not specify if 'surround' means 26 degree or per-neuron surround.

In the revised version of the manuscript, we now always clearly state which surround size we refer to.

L641: The description of surround in Methods L 641 is not clear: isn't the slope at the RF size equal to zero, by definition of RF size as the peak of the size tuning curve?

The analysis was applied to the size tuning curve beyond the RF. This has now been clarified in the text. See pp. 25-26.

L41 in the Abstract, I suggest changing 'small visual stimuli' to 'stimuli smaller than the RF'

We have implemented this change.

- Past work on surround modulation of variability. The Discussion text starting on L408 suggests that Festa et al. 2021 report no surround modulation of Fano factor. This is not correct, the main conclusion of that paper is that stimuli extending beyond the RF reduce the Fano factor, consistently with the predicted role of variability for probabilistic inference. Similar surround modulation of variability was also reported by Orban et al. 2016, which is worth discussing in relation to the findings presented here.

We apologize for some misunderstanding on our part of the data by Festa et al. We have now corrected the interpretation of the results of that study in the Discussion and also now cite Orban et al. 2016 (pp. 17, 19).

- Analysis of shared variance. The analysis of 'shared variance' applies factor analysis separately for each stimulus condition, but always assumes a single latent shared factor. This might not be the correct way to compare shared variance across conditions, because the latent dimensionality could change as well. The choice of dimensionality per condition should be based on cross-validation. The comparison could also be performed dimension per dimension, by comparing the amount of variance explained along each latent dimension.

We thank the Reviewer for this insightful comment. We have now made a more rigorous selection of the number of latent variables used in the analysis. To choose the latent space dimensionality, we performed 5-fold cross validation separately for each stimulus condition. For each cross-validation fold, we performed leave-neuron-out prediction for all neurons, and computed the root-mean-square error (RMSE) between the prediction and the recorded spike counts. The RMSE was then averaged over cross validation folds. In SG layers at most stimulus diameters, the RMSE decreased as the latent space dimensionality was increased from 1 to 2. However, the decrease was not statistically significant (using criterion t-test p < 0.05 RMSE at 1 vs 2-dimensional latent space with penetrations as observations). In the G layer, the RMSE vs latent space dimensionality curves were essentially flat. In IG layers, at all stimulus diameters, RMSE monotonically increased as the latent space dimensionality was increased. Thus, as there was not sufficient evidence in favor of increasing the latent space dimensionality, we kept the original dimensionality for the analysis (see Methods p. 26).

It’s good to keep in mind that the results of this analysis do not imply that V1 activity is one dimensional. It rather reflects the sample of V1 activity analyzed for this study. Because we used linear arrays with 100µm contact spacing, and as we factored our analyses based on layer, our sample comes from nearby neurons that at most are ~700 µm apart. To maximize the similarity of the receptive fields of the recorded neurons, we took great care to ensure that the recording probe was inserted vertically into the cortex. In fact, the orientation preferences of the neurons recorded in individual penetrations were nearly identical (see Figure 1 of Bijanzadeh et al. for an example penetration).

- Clarity.Methods inclusion criteria: state the fraction of neurons included ("visually responsive") out of the total channels recorded.

We now state in the manuscript that 82 of the 120 channels contained a visually responsive multi-unit. Please see Materials and methods under Experimental model (p. 22).

When averaging Fano factors, the geometric mean is more appropriate because FF is a ratio. When averaging changes across conditions, eg. Figure 2C,D, the median seems more appropriate given the outliers.

We followed the Reviewer’s suggestion and re-computed the average Fano-factor size tuning in Figure 2A and 2B using the geometric mean. We now use medians for Figure 2C-F. We have re-computed all the relevant values using medians and bootstrapped the statistical tests for median. This is specified in the results when describing these figures.

The error bars in 2F 3C 5E look wrong, not representative of the data distribution.

We have replaced standard deviations with bootstrapped confidence intervals in all relevant analyses. Please see Figures 2C-F, 3C-E, and 5 C-H.

Reviewer #2 (Recommendations for the authors):Cortical responses to sensory stimuli are variable, and that variability is often shared across a population of neurons. Variability and co-variability are likely to depend on where neurons sit in the cortical hierarchy and the relative weight of feedforward and recurrent connections. Stimulus size offers the opportunity to vary the relative weight of feedforward-excitatory (small stimuli) and recurrent excitatory/inhibitory (large stimuli) inputs to cortical neurons. The authors, therefore, set out to characterise the lamina-organisation of response variability and co-variability, and its dependence on stimulus size, in the primate visual cortex. They find while a stimulus usually suppresses variability and co-variability, both recover slightly at large stimulus sizes, at least for superficial layers of visual cortex.The measurements are derived from 5 penetrations of a 24-channel linear probe into V1 of 2 anaesthetised macaque monkeys. Multi-unit activity was identified on each channel and subject to analysis. Visual stimuli, usually consisting of simple patterns (gratings) of varying size were presented. The main strength of the work lies in using analyses of data obtained from primates, after careful assignment of recording sites to different layers of the cortex, to address interesting questions about whether processing in the visual cortex depends on which layer neurones are likely to belong to. The main weakness of the work lies in the fact that it relies on multi-unit activity, which is imperfect for the study of neural variability, particularly when variation in the stimulus (such as stimulus size) may result in the addition or removal of units from the MUA. Additionally, because there can be slow drifts in eye-position under anaesthesia, it is difficult to know whether fluctuations in response to small stimuli are due to changes in brain state or eye position.1. Increase in Fano Factor and shared variability for large stimuli. The authors conclude that variability increases as a stimulus grows larger, particularly in superficial layers. However, many units in superficial layers show complete surround suppression (e.g. Sceniak et al., 2001), and the MUA may be composed of different units when tested with small and large stimuli. This is presumably the case in the current data, as for large stimulus, the evoked rate is apparently no different to the baseline rate on average (Figure 3A), so the stimulus does not recruit an increase in mean activity over baseline, across the population of sites studied, and the Fano factors are derived from units that, on average, are not visually responsive. It, therefore, seems likely that Fano factors for large stimuli are derived from a different (or at least only partially overlapping) pool of neurones to that for smaller stimuli. I am not sure how to address this concern with MUA activity, because of the inherent ambiguity.

We have now performed the main analyses also for spike-sorted single units (SUs). The SU analysis is now shown in Figure 2—figure supplement 3. As we had just one visually responsive unit in the G layer, the analysis was performed only for SG and IG layers. While the SU data analysis is somewhat less robust than the analysis performed for multi-units (MUs; in Figure 2), due to the smaller sample size, the SU analysis shows similar results as the MU analysis. In particular, with respect to the Reviewer’s comment above, we still find that for many SUs in the SG layers, stimulation of the far RF-surround increased variability (Figure 2 Suppl 3, panels A, B and D). This indicates that the increase in Fano-factor for large stimuli involving the far surround was not caused by a change in the composition of neurons contributing to the measured MUA. Moreover, our SU analysis, just as the MU analysis, shows that depending on the SU, stimulation of the RF surround could increase, decrease, or not affect Fano-factor. The results of this analysis are described in the Supplementary Methods and Results.

2. Amplification of variability for small stimuli. The authors conclude that small stimuli increase variability. If true, this would be of high interest. The concern here is that trial-by-trial variability in mean rate might be confounded with small changes (including slow drifts) in eye-position, which are known to occur even under paralysis unless the eyes are restrained by other means (Forte et al., 2002). This could have different effects on variability as a function of stimulus size, as small but not larger stimuli move into and out of the receptive fields of some or all of the underlying units in the MUA during drift. For example, responses to the 0.1 deg stimulus in Figure 3A may be consistent with drift in the middle third of trials (this data also looks like it is a subset of that in Figure 4A-C, which shows the same pattern of effect for the smallest stimulus). The authors state on line 573: "To monitor eye movements, the RFs were remapped by hand approximately every 10-20 minutes and stimuli re-centered on the RF if necessary." To allow better comparison on such an important point it would be useful to include rasters for the smallest (e.g. 0.2 deg) stimuli giving a clear response in Figure 1. Additionally, given that 0.1 deg stimuli don't seem to elicit a response above baseline (Figure 2A) on average, it would seem better to represent 0.2 deg responses in Figure 2B. In general, however, without a secondary source of evidence that eye-position was stable for the duration of recording, it is difficult to know how to address these findings.

Because the RF of the neurons in our sample were precisely aligned retinotopically (see Bijanzadeh et al. 2018 for examples and detailed analysis), a shift in eye-position that could explain the amplification is expected to abolish neural responses across the whole array. To demonstrate that variability amplification can happen independent of eye movements, we show in Figure 3-suppl1 an example penetration that showed variability amplification for a 0.6deg stimulus, with no indication that eye movements may have occurred during that recording.

The work could be of interest because it makes measurements from the primate visual cortex and similar measurements have helped understand both vision and more general aspects of cortical function such as lamina-organisation, and the origins and impact of neural variability. Currently, however, the measurements do not provide highly convincing evidence for the conclusions that are drawn.3. I think these recordings are a subset of those in Bijanzadeh et al., (2018), which is referenced frequently and used similar lamina measurements (from 4 animals) but I am not sure. Please make this explicit. In addition, please clarify how the penetrations from each of the two animals are distributed (e.g. 2 in one animal, 3 in another). Finally, while it might be that statistical tests on the sites from each individual may not be informative, it would be of interest to know whether the effect sizes/directions were the same in each.

The currently presented data is indeed a subset of the data in Bijanzadeh et al. Here, we analyzed those penetrations in which stimulus presentation was repeated at least 20 times. Three of the penetrations were recorded in case MK366, and two in case MK374. We have added this information to the Methods (p. 22). Where feasible, we have plotted data for the two animals with different symbols (Figure 2 C-H, Figure 2-suppl3C-D, Figure 5C-H), and in Figure 2-suppl1 we show data from each penetration separately.

4. Spiking activity is defined as multiunit on line 107, but there are several occasions subsequently where words like 'simultaneously recorded units' (line 112), 'example units' (e.g. line 121), 'unit-by-unit' (line 174), 'majority of units' (line 175) are used, where the use of the term unit usually invokes the sense that a single neuron is been discussed. Indeed, on line 303 and in the legend to some Figs, the word 'neuron' is used, or on line 426 '15% of cells' is used. I think 'site' instead of 'unit' or 'neuron' or 'cells' needs to be used throughout, to avoid confusion.

Thank you for pointing out these discrepancies. We now specify throughout the manuscript whether we refer to single or multi-units.

5. The stimulus parameters were chosen to maximise the response of as many sites as possible on the relevant penetration. As there were only 5 sessions it would be good to record those parameters, and perhaps to make sure that the choice did not influence the outcomes (e.g. were particular penetrations associated with particular distributions of variance / co-variance, where that may have been because of the choice of stimulus?).

We now show the size tuning of Fano-factor for each penetration separately in Figure 2 —figure supplement 1.

6. The statement that the stimulus size yielding max quenching differs between layers (lines 188-9) should be accompanied by an appropriate statistical test or moderated.

In the revised manuscript, all layer comparisons are supported by statistical analysis or statements about comparisons were removed.

7. The mean matching conducted in Supplementary Figure seems key, and it deserves inclusion in the main manuscript. To be honest, I have read the figure description several times and am still unclear about what each of the panels shows.

The revised version of the manuscript includes an improved description of the mean-matching procedure (Figure 2-suppl 2) and the Results of this analysis are described in detail in the Supplementary Results (p. 39). Given how busy this figure is, we opted to keep it in the Supplementary Material, but we discuss it in both the main Results (p.10) and in the Supplementary Results (p. 39).

8. I cannot find anywhere the number of trials presented. This is important for understanding the robustness of the variability and particularly co-variability estimates. Must include explicitly, and also around discussion of mean-matching (ie. how many trials are retained in the analysis).

Depending on the penetration, 21-51 trials per stimulus were presented. The number of trials is now included in the manuscript (p. 22).

9. It is not clear to me why only sites with at least 5% surround suppression were studied (line 552). This would seem to be likely to bias the results, and perhaps differently in different layers given the generally stronger suppression found in superficial layers. What happens if that filter is removed?

We have now removed this filter from our analysis for both MUs and SUs. As our dataset did not contain MUs with lower than 5% suppression (the filter was a relic from previous iterations of the manuscript), changing the filter did not change the multi-unit analysis presented in the paper.

[Editors’ note: what follows is the authors’ response to the second round of review.]

The manuscript has been improved but there are some remaining issues that need to be addressed, as detailed below in the comments from Reviewer 3:Reviewer #3 (Recommendations for the authors):I appreciate the effort that the authors have put into the reply, and very much understand the delays that normal life imposes. I apologise similarly that it has taken me this long to analyse the responses and revised manuscript. As in my initial assessment, I continue to believe that the main strength of the work lies in using analyses of data obtained from primates, after careful assignment of recording sites to different layers of the cortex, to address questions about layer-dependence of processing in the visual cortex.The authors have addressed many of my concerns, but while the new analyses of single-unit activity show that variability can depend on size in single-units (including some which show increase in variability for larger sizes) they do not convincingly demonstrate a layer dependence of these effects. Further, even in multi-unit activity, this larger-size-effect appears to be strongest in one or two of the recordings, and weak if present in other recordings. I am therefore not sure that a major conclusion of the abstract ("variability is tuned for stimulus size in a layer-dependent manner.") can be robustly supported by the current data.In the following I address the authors replies that I remain concerned about:1. The authors state: "… While the SU data analysis is somewhat less robust than the analysis performed for multi-units (MUs; in Figure 2), due to the smaller sample size, the SU analysis shows similar results as the MU analysis."Thank you for performing these analyses and including the new data. However, while Figure 2 —figure supplement 3 shows individual units in which variability can increase with stimulus size, on average (panel B of this supplementary data) neither SG nor IG show an increase in Fano Factor at larger sizes. I therefore think that this statement rests on the observation that many of the units in Panel D (which shows relative size at which there is peak facilitation) show values larger than 1 in SG but fewer are larger than 1 in IG. I would appreciate clearer description of this dependency in the main text, and more stringent tests of the hypothesis that in SU as in MU, variability is greater at larger sizes, particularly in SG.

We pointed out in the earlier version of the manuscript (p.9 lines 246-252; p9 second paragraph in the revised version) that individual units showing increases, decreases or no effect of surround stimulation on Fano factor are found in all layers, but what varies is the relative proportion of these units in the different layers. Our SU analysis is inevitably less robust because of the small sample size that reduces significant differences. However, the trend is clearly similar to the trend seen for MUA, i.e. in Figure 2 Suppl 3 panel D, there are more units in SG layers showing facilitation at larger sizes than in IG layers, but due to the small sample, the increase in Fano Factor (FF) at the largest size in panel B does not reach statistical significance. Note also, that the difference in the analysis between panel B and D is that in panel B we analyze FF at pre-chosen stimulus sizes for all cells and then average at those sizes across cells; instead, in panel D, we ask for each individual cell, at what size is max FF observed? The analysis in panel D is, thus, more sensitive because it looks for increases in FF at ANY stimulus size. The point of the SU analysis in the first revision of the manuscript was to address the Reviewer’s concern that increases in FF for large stimuli may be an artifact of the MUA analysis. In the Revision we showed this concern is invalid, as we see this phenomenon in the responses of single cells. We have revised the text to clarify all this (see p.11 2nd paragraph).

In addition, I believe that reanalyses are required here to ensure that the conclusions are correct -the Supplementary Methods state that 28 SG and 17 IG SUs were included in the analyses (and 7/8 respectively were excluded because the tuning curve for Fano Factor was 'noisy and essentially flat'), and I count 28 SG and 16 IG data points (one may be obscured) in panel B. However I count 31 SG and 24 IG data points in panel C (some may be obscured, and there is too much overlap in panel D to count them). I am therefore concerned that the analyses may have inadvertently included units that should not be included, because their Fano Factors were not adequately tuned to measure the size at the maximum or minimum.

Thank you for pointing out this mistake. We now use the correct sample of units in panels C and D of Figure 2-Suppl3, consistent with the number of units used for the analysis in panel B (28 SG and 17 IG units), and have corrected the numbers and statistics in the text where appropriate see new panels C,D in Figure 2—figure supplement 3, and Supplementary Results (*Single unit analysis*). Following the Reviewer’s suggestion, we have additionally performed new statistical tests for the data in these panels, and now show that the increase in FF for large stimuli in SG layers (panel D) is statistically significant, while the increase in FF for the IG layers is not significant (see Supplementary Results, *Single unit analysis*). These corrections have strengthened our results and conclusions.

2. The authors state: "Because the RF of the neurons in our sample were precisely aligned retinotopically (see Bijanzadeh et al. 2018 for examples and detailed analysis), a shift in eye-position that could explain the amplification is expected to abolish neural responses across the whole array. To demonstrate that variability amplification can happen independent of eye movements, we show in Figure 3-suppl1 an example penetration that showed variability amplification for a 0.6deg stimulus, with no indication that eye movements may have occurred during that recording."Thank you for the new figure. I am not sure how to read Figure 3-suppl1, however, as it shows rasters of MU activity for different trials but does not quantify variability among those or the amplification of variability – the figure needs accompanying quantitative analysis to be able to interpret and justify the statement in results that "Figure 3—figure supplement 1 rules out eye movements as a potential of increased variability for stimuli smaller than the RF size".

We have modified Figure 3-figure suppl1. Each panel in Figure 3-figure suppl1A, shows the response of 17 single units across the array spanning all layers, in a single trial, in which a 0.6° stimulus was presented. In total 50 trials are shown for these same units. In Figure 3-figure suppl1B, we now show the PSTH of Fano-factor for 5 units showing variability amplification in this penetration and stimulus condition. The unit # above each panel corresponds to the unit # on the Y axis of the raster plots. One additional unit from this case is also shown in Figure 3A (left column). If an eye movement had occurred, it would be expected to abolish the responses across the whole array. In none of the panels in (A) this is observed, suggesting the variability amplification seen in the 5 units in panel (B) is due to eye movements. The figure is explained in the figure legend and discussed in the Results (p. 13 bottom paragraph).

In addition, the influence of small eye movements on trial-by-trial measures of variability will presumably depend on receptive field size, which varies with layer. Influence of eye-movements on variability as a function of time into trial may also depend on phase-sensitivity (i.e. simple-cell like), which can also vary with layer. I don't want to push on this too much, however I do think that the potential influence of small eye-movements/drift should be discussed as a limitation.

In Figure 3-figure suppl1, we present raster plots in response to a small stimulus (0.6 deg), precisely because it is more prone to artifacts due to eye movements. We show that in this example penetration, we see increases in FF for this stimulus, relative to baseline, in the absence of eye movements. In all honesty, we could not with certainty discount eye movements in all penetrations as a potential cause of increases in FF for small stimuli. But the fact that we do see this effect in cases in which we can confidently discard eye movements as possible cause speaks for the reality of the phenomenon. Nevertheless, following the Reviewer’s suggestion, we now discuss the potential limitation of eye movements in the Discussion section (see p. 21 end of top paragraph).

3. The authors state: "Where feasible, we have plotted data for the two animals with different symbols (Figure 2 C-H, Figure 2-suppl3C-D, Figure 5C-H), and in Figure 2-suppl1 we show data from each penetration separately."

The data presented in Figure 2-figure suppl1 (now panel A) shows population-averaged FF-size tuning curves across all cells in a penetration (grouped by layer). This averaging diminishes effects if the FF increase occurs at slightly different stimulus sizes in different cells. We have now added to this supplementary figure a plot of Figure 2G and 2H with individual cells color coded by penetration. It is clear from these new plots, that all penetrations show increases in FF for large stimuli in all layers, but more so in the SG layers. We did the same for the SU analysis, where we now plot individual cells color-coded by penetration number in panels C and D of new Figure 2-figure suppl3. We hope this is a sufficiently convincing demonstration that the facilitation of FF in SG layers is not just seen in a single penetration. Performing statistical analysis on individual penetrations is not appropriate here as the number of cells in each penetration is insufficient for statistical power. We refer to these new plots in the Results (see p.9, 3^st^ and 4^nd^ paragraphs).

Thank you for distinguishing the animals in the figures and for including the data for individual recordings in Figure 2 -suppl1. I may be missing something, but my reading of the panels in this supplementary data is that while Fano factor decreases from smallest to larger sizes in most or all penetrations, subsequent increase of Fano factors at larger sizes is strongest in the penetration denoted by red lines, where it is present in all layers but particularly prominent in SG. It is hard to see clear evidence for larger-size-dependence outside this penetration. I therefore think that it is important to perform statistical analyses that take into account the particular penetration on which sites were recorded in assessing layer differences in the increase in Fano Factor at larger sizes. This is likely to be necessary for SU as well as MU analyses.Other points arising from revisions:1. Figure 1: Are panels Figure 1B and Figure 1C flipped? The text associated with these on page 6 is more consistent with such an arrangement: "…increasing the stimulus diameter beyond the RF boundaries did not affect Fano-factor for the G layer MU (Figure 1B)", "but for larger stimuli (involving the "far" RF surround), Fano factor increased back to approximately its value for a stimulus matched to the RF size (Figure 1C)".

The figures are correct. To clarify this we now point out that the RF size was 0.5 deg for these units (see p. 6, 2^nd^ paragraph).

2. Page 7: "In contrast, in IG layers, the minimum Fano-factor was reached at a stimulus diameter approximately twice the RF diameter (we term this the "near surround"), and beyond this stimulus size, Fano-factor increased back to its value for a stimulus matched to the RF diameter (Figure 2A Right).". It is hard to see this in Figure 2a. Perhaps an arrow to indicate the position of the minimum.

We have added two arrows in Figure 2A, one pointing to the RF size, the other to the minimum FF value, and corrected the figure legend accordingly. We also slightly modified the description of Figure 2A in the Results (see p. 7 top paragraph).

3. Page 7: "In G layers, Fano-factor reached its minimum for a stimulus the size of the RF diameter" While the change from 1x to 2x is only significant in IG, both IG and G reach a minimum at 2xRF.

We changed this sentence as follows: “In G layers, Fano-factor decreased as a small stimulus was increased to fill the RF size, and did not change significantly for larger stimuli (one sample t-test p > 0.19).” (see p. 7 bottom paragraph).